# TP53 copy number expansion is associated with the evolution of increased body size and an enhanced DNA damage response in elephants

Michael Sulak[1], Lindsey Fong[1], Katelyn Mika[1], Sravanthi Chigurupati[1], Lisa Yon[2,3], Nigel P Mongan[2,3,4], Richard D Emes[2,3,5], Vincent J Lynch[1]*

[1]Department of Human Genetics, The University of Chicago, Chicago, United States; [2]School of Veterinary Medicine and Science, University of Nottingham, Leicestershire, United Kingdom; [3]Faculty of Medicine and Health Sciences, University of Nottingham, Leicestershire, United Kingdom; [4]Department of Pharmacology, Weill Cornell Medical College, New York, United States; [5]Advanced Data Analysis Centre, University of Nottingham UK, Nottingham, United Kingdom

*For correspondence: vjlynch@uchicago.edu

Competing interests: The authors declare that no competing interests exist.

**Abstract** A major constraint on the evolution of large body sizes in animals is an increased risk of developing cancer. There is no correlation, however, between body size and cancer risk. This lack of correlation is often referred to as 'Peto's Paradox'. Here, we show that the elephant genome encodes 20 copies of the tumor suppressor gene *TP53* and that the increase in *TP53* copy number occurred coincident with the evolution of large body sizes, the evolution of extreme sensitivity to genotoxic stress, and a hyperactive TP53 signaling pathway in the elephant (Proboscidean) lineage. Furthermore, we show that several of the *TP53* retrogenes (*TP53RTGs*) are transcribed and likely translated. While *TP53RTGs* do not appear to directly function as transcription factors, they do contribute to the enhanced sensitivity of elephant cells to DNA damage and the induction of apoptosis by regulating activity of the TP53 signaling pathway. These results suggest that an increase in the copy number of *TP53* may have played a direct role in the evolution of very large body sizes and the resolution of Peto's paradox in Proboscideans.

## Introduction

Lifespan and maximum adult body size are fundamental life history traits that vary considerably between species (*Healy et al., 2014*). The maximum lifespan among vertebrates, for example, ranges from over 211 years in the bowhead whale (*Balaena mysticetus*) to only 59 days in the pygmy goby (*Eviota sigillata*) whereas body sizes ranges from 136,000 kg in the blue whale (*Balaenoptera musculus*) to 0.5 g in the Eastern red-backed salamander (*Plethodon cinereus*) (*Healy et al., 2014*). Similar to other life history traits, such as body size and metabolic rate or body size and age at maturation, body size and lifespan are strongly correlated such that larger species tend to live longer than smaller species (*Figure 1A*). While abiotic and biological factors have been proposed as major drivers of maximum body size evolution in animals, maximum body size within tetrapods appears to be largely determined by biology (*Smith et al., 2010*; *Sookias et al., 2012*). Mammals, for example, likely share biological constraints on the evolution of very large body sizes with rare breaks in those constraints underlying the evolution of gigantism in some lineages (*Sookias et al., 2012*), such as Proboscideans (elephants and their and extinct relatives), Cetaceans (whales), and the extinct hornless rhinoceros *Paraceratherium* ('Walter').

**eLife digest** As time passes, healthy cells are more likely to become cancerous because more and more damaging mutations accumulate in the cell's DNA. Assuming that all cells have a similar risk of acquiring mutations, larger and longer-lived animals – like elephants – should have a higher risk of cancer than smaller, shorter-lived animals – like mice. However, there does not appear to be any link between the size of an animal and its risk of developing cancer. Consequently, a key question in cancer biology is how very large animals protect themselves against these diseases.

One gene that is often damaged during an animal's lifetime is called *TP53*. This gene normally produces a tumor suppressor protein that senses when DNA is damaged or a cell is under stress and either briefly slows the cell's growth while the damage is repaired or triggers cell death if the stress is overwhelming. One way that large animals could reduce their risk of cancer is to have extra copies of *TP53* or other genes that encode tumor suppressor proteins.

Here Sulak et al. used an evolutionary genomics approach to study *TP53* in 61 animals of various sizes, including several large animals such as African elephants and Minke whales. All of the animals studied had at least one copy of *TP53*, and several had a few extra copies, known as *TP53* retrogenes. African elephants – the largest living land mammal – had more retrogenes than any of the others with 19 in total. To investigate why African elephants have so many TP53 retrogenes, Sulak et al. also analyzed DNA from Asian elephants and several other closely related, but now extinct species, including the woolly mammoth. As expected, as species evolved larger body sizes they also evolved more *TP53* retrogenes.

Further experiments indicate that several of the *TP53* retrogenes in African elephants are likely to be able to produce the tumor suppressor protein and that they contribute to elephant cells being better equipped to deal with DNA damage. The next step following on from this work will be to find out exactly how *TP53* retrogenes help to protect animals from cancer.

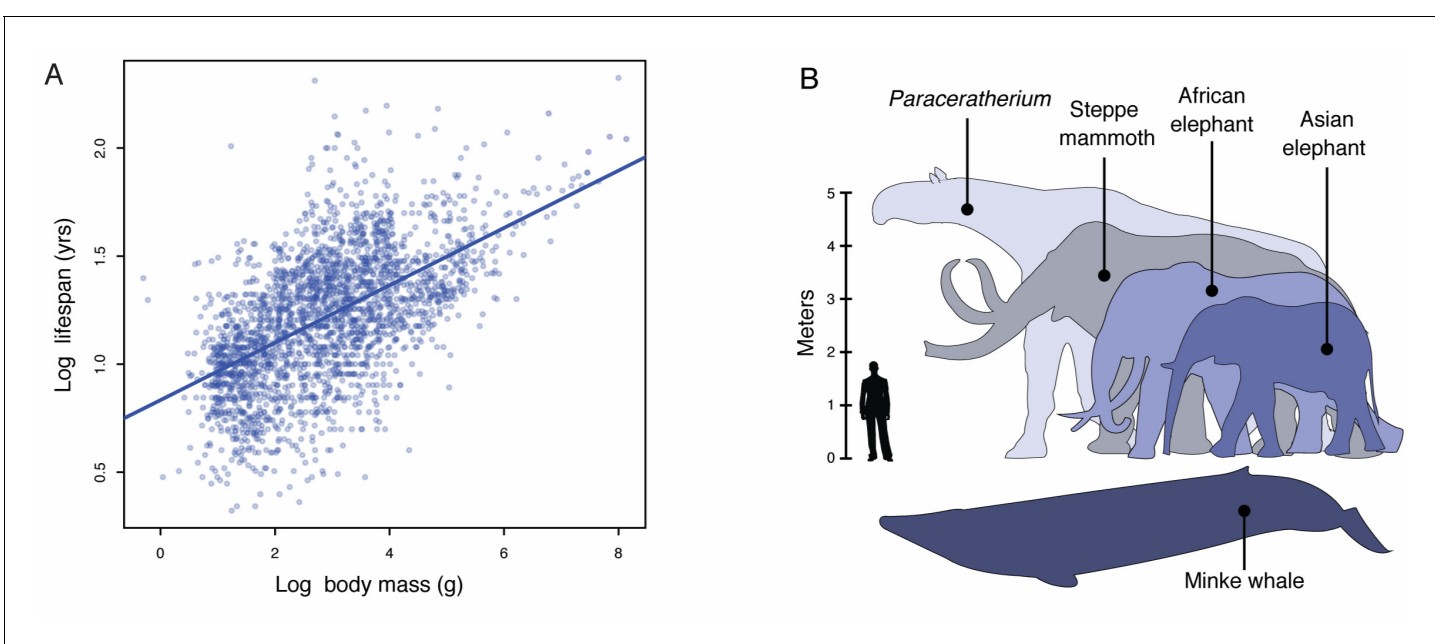

**Figure 1.** Body size evolution in vertebrates. (**A**) Relationship between body mass (g) and lifespan (years) among 2556 vertebrates. Blue line shows the linear regression between log (body mass) and log (lifespan), $R^2$ = 0.32. (**B**) Body size comparison between living (African and Asian elephants) and extinct (Steppe mammoth) Proboscideans, Cetaceans (Minke whale), and the extinct hornless rhinoceros *Paraceratherium* ('Walter'), and humans.

A major constraint on the evolution of large body sizes in animals is an increased risk of developing cancer. If all cells have a similar risk of malignant transformation and equivalent cancer suppression mechanisms, organism with many cells should have a higher risk of developing cancer than organisms with fewer cells; Similarly organisms with long lifespans have more time to accumulate cancer-causing mutations than organisms with shorter lifespans and therefore should be at an increased risk of developing cancer, a risk that is compounded in large bodied, long-lived organisms (*Cairns, 1975*; *Caulin and Maley, 2011*; *Doll, 1971*; *Peto, 2015*; *Peto et al., 1975*). There are no correlations, however, between body size and cancer risk or lifespan and cancer risk across species (*Leroi et al., 2003*), this lack of correlation is often referred to as 'Peto's Paradox' (*Caulin and Maley, 2011*; *Peto et al., 1975*). Epidemiological studies in wild populations of Swedish roe deer (*Capreolus capreolus*) and beluga whales (*Delphinapterus leucas*) in the highly polluted St. Lawrence estuary, for example, found cancer accounted for only 2% (*Aguirre et al., 1999*) and 27% (*Martineau et al., 2002*) of mortality, respectively, much lower than expected given body size of these species (*Caulin and Maley, 2011*).

Among the mechanisms large, long lived animals may have evolved that resolve Peto's paradox are a decrease in the copy number of oncogenes, an increase in the copy number of tumor suppressor genes (*Caulin and Maley, 2011*; *Leroi et al., 2003*; *Nunney, 1999*), reduced metabolic rates leading to decreased free radical production, reduced retroviral activity and load (*Katzourakis et al., 2014*), increased immune surveillance, and selection for 'cheater' tumors that parasitize the growth of other tumors (*Nagy et al., 2007*), among many others. Naked mole rats (*Heterocephalus glaber*), for example, which have very long lifespans for a small-bodied organism evolved cells with extremely sensitive contact inhibition likely acting as a constraint on tumor growth and metastasis (*Seluanov et al., 2009*; *Tian et al., 2013*). Similarly long-lived blind mole rats (*Splanx sp.*) evolved an enhanced *TP53*-signaling and necrotic cell death mechanisms that also likely constrains tumor growth (*Ashur-Fabian et al., 2004*; *Avivi et al., 2007*; *Avivi et al., 2005*; *Gorbunova et al., 2012*; *Manov et al., 2013*). Thus, while some of the mechanisms that underlie cancer resistance in small, long-lived mammals have been identified, the mechanisms by which large bodied animals evolved enhanced cancer resistance are unknown.

Here we use evolutionary genomics and comparative cell biology to explore the mechanisms by which elephants, the largest extant land mammal (*Figure 1B*), have evolved enhanced resistance to cancer. We found that the elephant genome encodes a single *TP53* gene and 19 *TP53* retrogenes, several of which are transcribed and translated in elephant tissues. Comparison of the African and Asian elephant TP53 gene copy number with the copy number in the genome of the extinct American mastodon, woolly mammoth, and Columbian mammoth indicates that copy number increased relatively rapidly coincident with the evolution of large body-sizes in the Proboscidean lineage. Finally, we show that elephant cells have an enhanced response to DNA-damage that is mediated by a hyperactive TP53 signaling pathway and that this augmented TP53 signaling is dependent upon TP53 retrogenes and can be transferred to the cells of other species through exogenous expression of elephant TP53 retrogenes. These results suggest that the origin of large body sizes, long lifespans, and enhanced cancer resistance in the elephant lineage evolved at least in part through reinforcing the anti-cancer mechanisms of the major 'guardian of the genome' TP53.

## Results

### Expansion of the *TP53* repertoire in proboscideans

We characterized *TP53* copy number in 61 Sarcopterygians (Lobe-finned fishes) with draft or completed genomes, including large, long-lived mammals such as the African elephant (*Loxodonta africana*), Bowhead (*Balaena mysticetus*) and Minke (*Balaenoptera acutorostrata scammoni*) whales. We found that all Sarcopterygian genomes encoded a single *TP53* gene and that some lineages also contained a few *TP53* retrogenes (*TP53RTG*), including marsupials, Yangochiropteran bats, and Glires, in which 'processed' *TP53* pseudogenes have previously been reported (*Ciotta et al., 1995*; *Czosnek et al., 1984*; *Hulla, 1992*; *Tanooka et al., 1995*; *Weghorst et al., 1995*; *Zakut-Houri et al., 1983*). We also identified a single *TP53RTG* gene in the lesser hedgehog tenrec (*Echinops telfairi*), which had been previously reported (*Belyi et al., 2010*), rock hyrax (*Procavia capensis*),

and West Indian manatee (*Trichechus manatus*). The African elephant genome, however, encoded 19 TP53RTG genes (*Figure 2A*), 14 of which retain potential to encode truncated proteins (*Table 1*).

To trace the expansion of TP53RTG gene family in the Proboscidean lineage with greater phylogenetic resolution, we used three methods to estimate the minimum (1:1 orthology), average (normalized read depth), and maximum (gene tree reconciliation) *TP53/TP53RTG* copy number in the Asian elephant (*Elephas maximus*), extinct woolly (*Mammuthus primigenius*) and Columbian (*Mammuthus columbi*) mammoths, and the extinct American mastodon (*Mammut americanum*) using

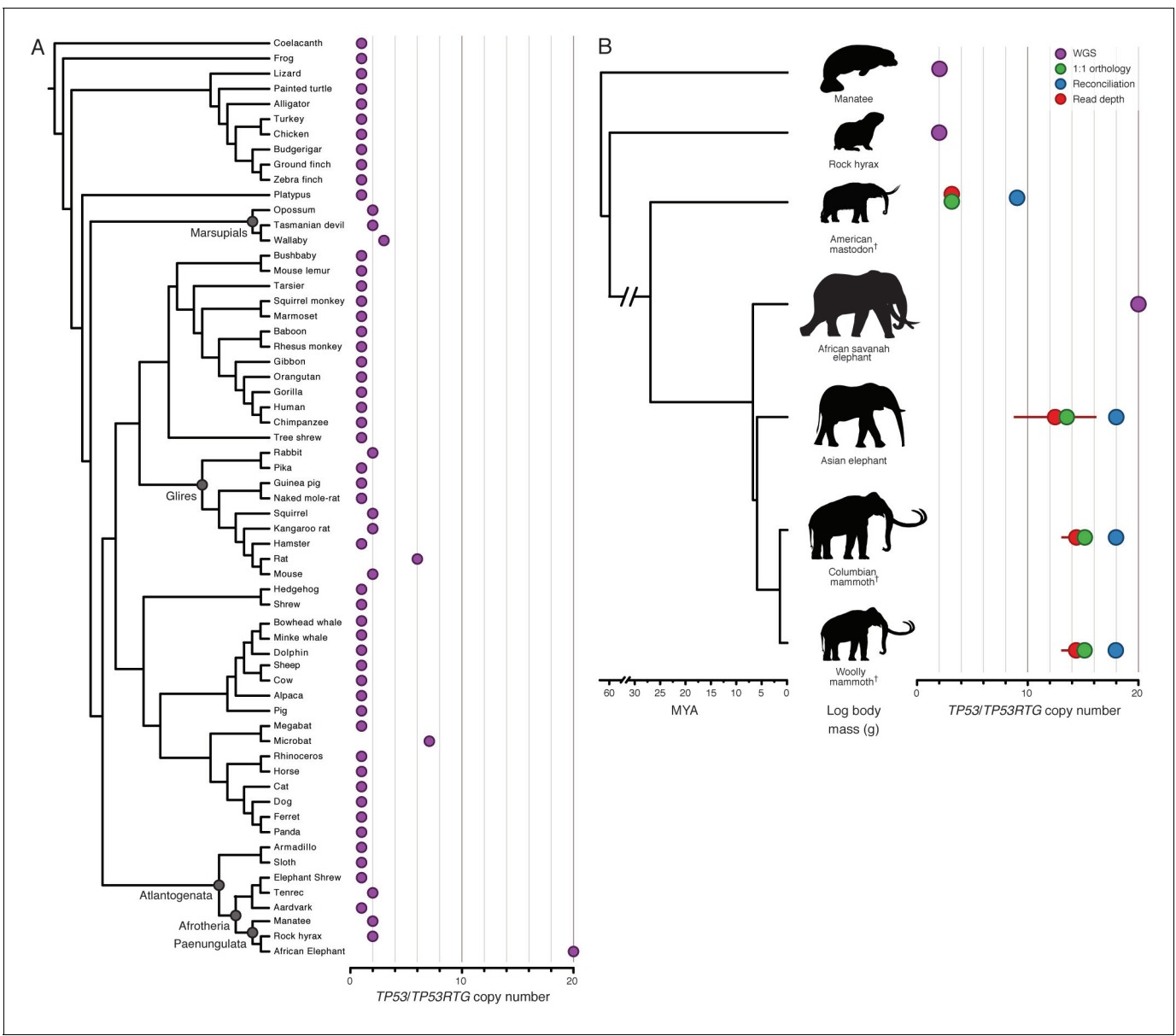

**Figure 2.** Expansion of the TP53RTG gene repertoire in Proboscideans. (A) TP53 copy number in 61 Sarcopterygian (Lobe-finned fish) genomes. Clade names are shown for lineages in which the genome encodes more than one TP53 gene or pseudogene. (B) Estimated *TP53/TP53RTG* copy number inferred from complete genome sequencing data (WGS, purple), 1:1 orthology (green), gene tree reconciliation (blue), and normalized read depth from genome sequencing data (red). Whiskers on normalized read depth copy number estimates show the 95% confidence interval of the estimate.

The following figure supplement is available for figure 2:

**Figure supplement 1.** Reconciled *TP53/TP53RTG* gene trees.

**Table 1.** Summary information for African elephant *TP53/TP53RTG* genes.

| Gene Id | ENSEMBL Id | Scaffold (loxAfr3) | Chromosome (loxAfr4) | Coding potential | ORF size |
|---------|-----------|--------------------|----------------------|------------------|----------|
| *TP53* | ENSLAFG00000007483 | 47 | Chr 11 | Yes | 392aa |
| *TP53RTG1* | ENSLAFG00000025553 | 175 | Unmapped | No | N/A |
| *TP53RTG2* | N/A | 217 | Unmapped | Yes | 134aa |
| *TP53RTG3* | ENSLAFG00000027474 | 406 | Unmapped | Yes | 79aa |
| *TP53RTG4* | N/A | 627 | Unmapped | Yes | 134aa |
| *TP53RTG5* | ENSLAFG00000027348 | 221 | Unmapped | Yes | 162aa |
| *TP53RTG6* | N/A | 76 | Chr 27 | Yes | 123aa |
| *TP53RTG7* | N/A | 208 | Unmapped | No | N/A |
| *TP53RTG8* | ENSLAFG00000027820 | 294 | Unmapped | Yes | 210aa |
| *TP53RTG9* | ENSLAFG00000027669 | 786 | Unmapped | No | N/A |
| *TP53RTG10* | ENSLAFG00000030555 | 221 | Unmapped | Yes | 210aa |
| *TP53RTG11* | N/A | 281 | | Yes | 203aa |
| *TP53RTG12* | ENSLAFG00000028299 | 825 | Unmapped | Yes | 180aa |
| *TP53RTG13* | ENSLAFG00000032042 | 458 | Unmapped | No | N/A |
| *TP53RTG14* | ENSLAFG00000026238 | 928 | Unmapped | Yes | 210aa |
| *TP53RTG15* | ENSLAFG00000027365 | 656 | Unmapped | Yes | 210aa |
| *TP53RTG16* | ENSLAFG00000030880 | 378 | Unmapped | No | N/A |
| *TP53RTG17* | ENSLAFG00000028692 | 552 | Unmapped | Yes | 111aa |
| *TP53RTG18* | N/A | 498 | Unmapped | Yes | 111aa |
| *TP53RTG19* | ENSLAFG00000032258 | 342 | Unmapped | Yes | 210aa |

existing whole genome sequencing data (*Enk et al., 2014*, *2013*; *Rohland et al., 2010*; *Wilkie et al., 2013*). As expected, we identified a single canonical TP53 gene in these species and estimated the *TP53RTG* copy number in the Asian elephant genome to be 12–17, approximately 14 in both the Columbian and woolly mammoth genomes, and 3–8 in the 50,000–130,000 year old American mastodon genome (*Figure 2B* and *Figure 2—figure supplement 1*). These data indicate that large-scale expansion of the *TP53RTG* gene family occurred in the Proboscidean lineage and suggest that *TP53RTG* copy number was lower in ancient Proboscideans such as the mastodon, which diverged from the elephant lineage ~ 25 MYA (*Rohland et al., 2010*), than in recent species such as elephants and mammoths.

## The *TP53RTG* repertoire expanded through repeated segmental duplications

Several mechanisms may have increased the *TP53RTG* copy number in the Proboscidean lineage including serial retrotransposition from the *TP53* gene, serial retrotransposition from the *TP53* and one or more daughter transcribed retrogenes, repeated segmental duplications of chromosomal loci containing *TP53RTG* genes, or some combination of these mechanisms. Consistent with copy number expansion through a single retrotransposition event followed by repeated rounds of segmental duplication, we found that each *TP53RTG* retrogene was flanked by nearly identical clusters of transposable elements (*Figure 3A*) and embedded within a large genomic region with greater than 80% sequence similarity (*Figure 3B*). Next we used progressiveMAUVE to align the 18 elephant contigs containing *TP53RTG* retrogenes and found that they were all embedded within large locally collinear blocks that span nearly the entire length of some contigs (*Figure 3C*), as expected for segmental duplications.

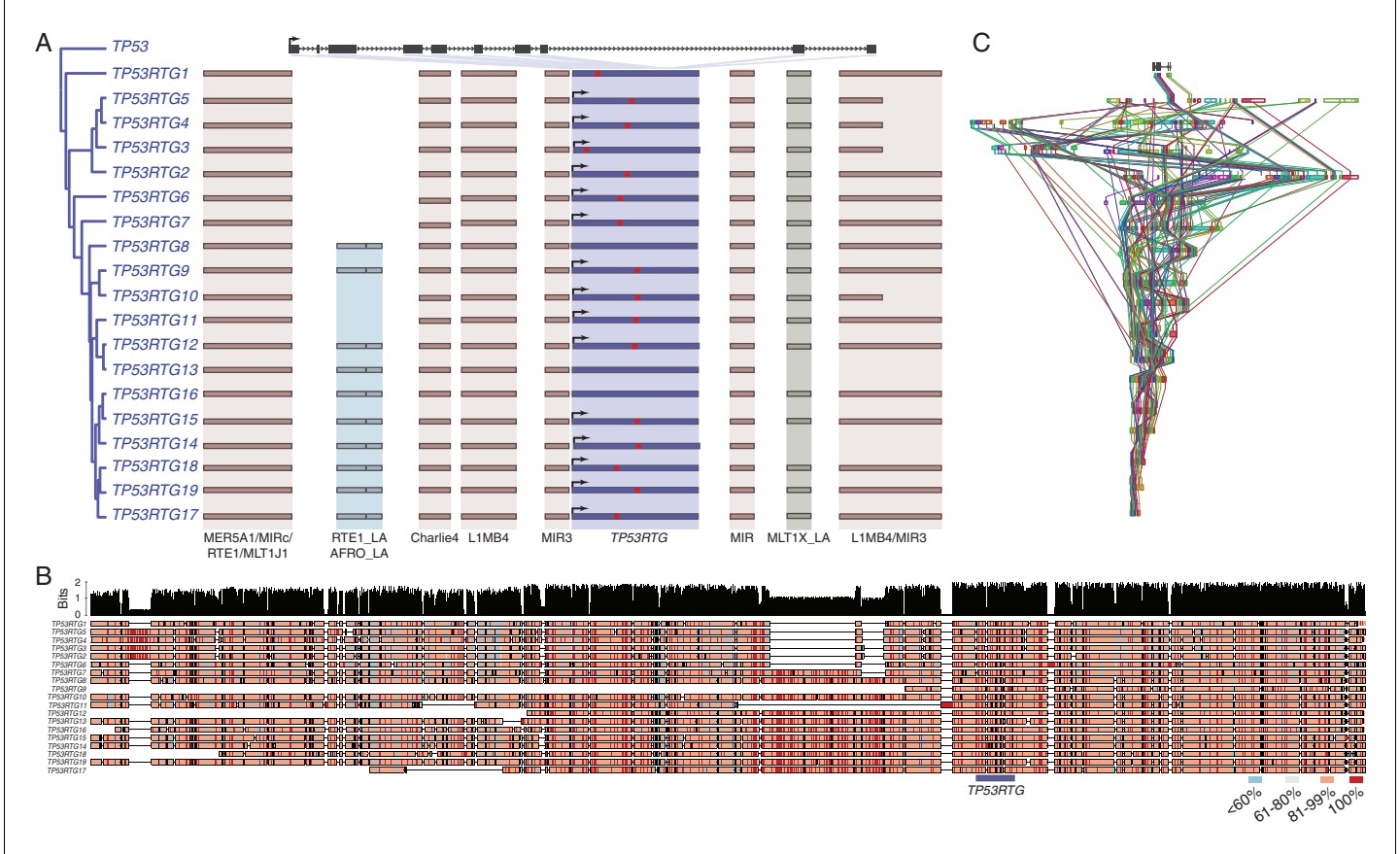

**Figure 3.** TP53RTG copy number increased through segmental duplications. (**A**) Organization of the TP53 and TP53RTG loci in African elephant. The *TP53/TP53RTG* gene tree is shown at the left. The location of homologous transposable elements that flank the TP53RTG genes are shown and l Abegglen ed. TP53RTG genes with intact start codons are labeled with arrows, stop codons are shown in red. (**B**) Multiple sequence alignment (MUSCLE) of elephant TP53RTG containing contigs. The location of the TP53RTG genes is shown with a blue bar. Sites are color coded according to their conservation (see inset key). (**C**) ProgressiveMAUVE alignment of elephant TP53RTG containing contigs. Colored boxes shown the location of collinear blocks, lines connect homologous collinear blocks on different contigs.

## *TP53RTG* copy number expansion is correlated with proboscidean body size

Our observation that *TP53RTG* genes expanded through segmental duplications suggests they may have a tree-like phylogenetic history that preserves information about when in the evolution of Proboscideans the duplicates occurred. Therefore we assembled a dataset of *TP53/TP53RTG* orthologs from 65 diverse mammals and jointly inferred the *TP53/TP53RTG* gene tree and duplication dates in a Bayesian framework to determine if *TP53/TP53RTG* copy number was correlated with body size evolution in Proboscideans. For comparison, we also inferred the phylogenetic history of *TP53/TP53RTG* genes using maximum likelihood and an additional Bayesian method. We found all phylogenetic inference methods inferred that the TP53RTG genes from elephant, hyrax, and manatee formed a well-supported sister clade to the canonical genes from these species, whereas the tenrec *TP53* and *TP53RTG* genes formed a separate well-supported clade (*Figure 4A* and *Figure 4—figure supplement 1*). These data indicate that retrotransposition of *TP53* occurred independently in tenrecs and in the elephant, hyrax, and manatee stem-lineage (Paenungulata), followed by expansion of *TP53RTG* genes in the Proboscidean lineage.

Based on our time-calibrated phylogeny, we inferred that the initial retrotransposition of the TP53 gene in the Paenungulata stem-lineage occurred approximately 64 MYA (95% HPD = 62.3–66.2 MYA; *Figure 4A*). This was followed by a period of ~25 million years during which no further retrotranspositions or segmental duplications were fixed in the genome, however, the TP53RTG

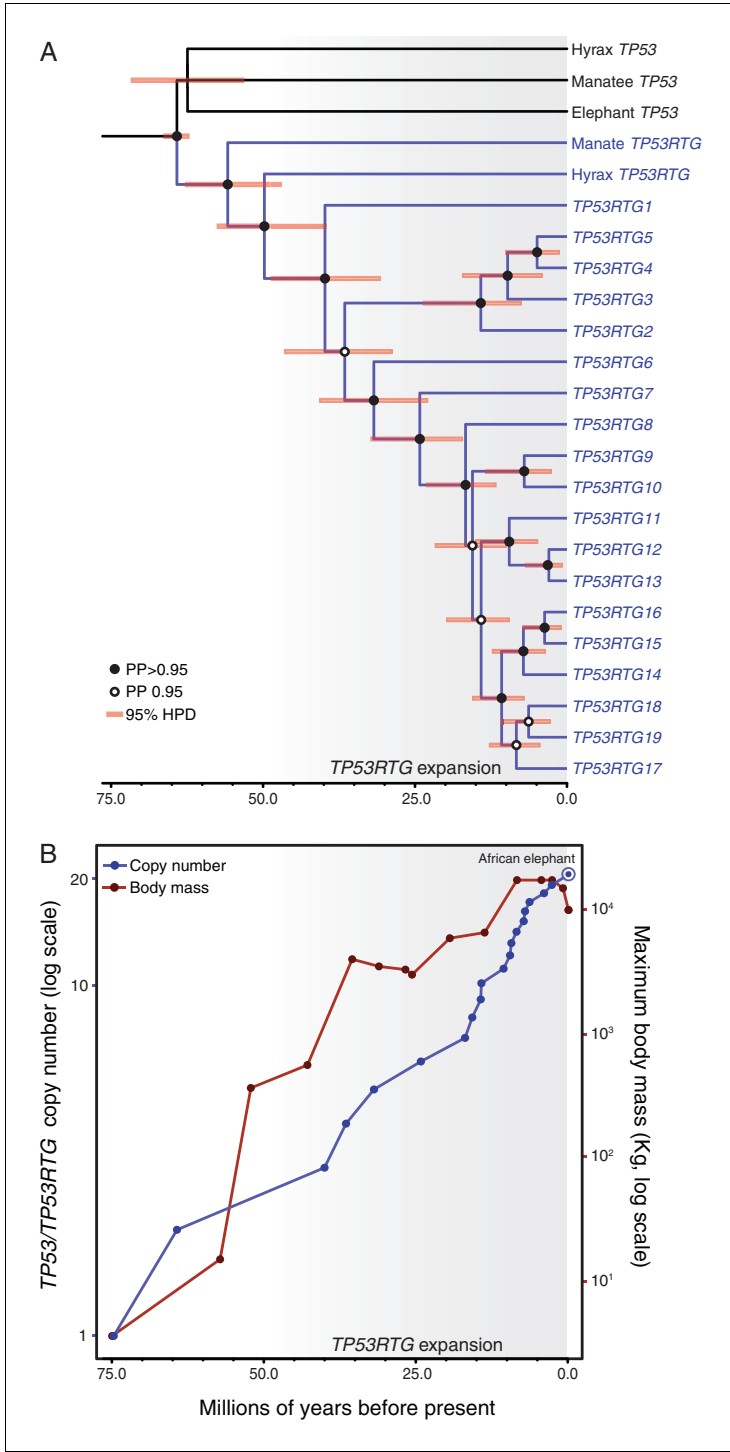

**Figure 4.** TP53RTG copy number is correlated with body size evolution in Proboscideans. (**A**) Time calibrated Bayesian phylogeny of *TP53/TP53RTG* genes. *TP53RTG* genes are shown in blue, the 95% highest posterior density (HPD) of estimated divergence dates are shown as red bars, nodes with a posterior probability (PP) (PP) > 0.95 are labeled with closed circles whereas nodes with a PP ≤ 0.95.95 are labeled with open circles. The period corresponding to the expansion of the *TP53RTG* gene repertoire is shown in a grey. (**B**) *TP53/TP53RTG* copy number (blue) and Proboscidean body size (red) increases through time are correlated.

The following figure supplement is available for figure 4:

**Figure supplement 1.** TP53/TP53RTG gene trees.

gene family rapidly expanded after ~40 MYA (95% HPD = 30.8–48.6 MYA; *Figure 4A*). To correlate *TP53/TP53RTG* copy number and the origin of large body sizes in Proboscideans we estimated *TP53/TP53RTG* copy number through time and gathered data on ancient Proboscidean body sizes from the literature (*Evans et al., 2012*; *Smith et al., 2010*). We found that the increase in *TP53/TP53RTG* copy number in the Proboscidean lineage and Proboscidean body size evolution closely mirrored each other (*Figure 4B*).

## *TP53RTG12* is transcribed from a transposable element derived promoter

If expansion of the *TP53RTG* gene repertoire played a role in the resolution of Peto's paradox during the evolution of large bodied Proboscideans, then one or more of the *TP53RTG* genes should be transcribed. Therefore we generated RNA-Seq data from Asian elephant dermal fibroblasts, African elephant term placental villus and adipose tissue, and used previously published RNA-Seq data from Asian elephant PBMCs (*Reddy et al., 2015*) and African elephant fibroblasts (*Cortez et al., 2014*) to determine if *TP53RTG* genes were transcribed. We found that the *TP53* and *TP53RTG12* genes were robustly transcribed in all samples, whereas *TP53RTG3* and *TP53RTG18* transcripts were much less abundant (*Figure 5A*). To confirm that the African and Asian elephant *TP53RTG* genes were transcribed, we designed a set of PCR primers specific to the *TP53* and *TP53RTG* genes that flank a diagnostic 15–30 bp deletion in *TP53RTG* genes (*Figure 5—figure supplement 1*) and used RT-PCR to assay for expression in Elephant fibroblast cDNA generated from DNase treated RNA. Consistent with transcription of the *TP53RTG* genes, we amplified PCR products at the expected size for the *TP53* and *TP53RTG* transcripts but did not amplify PCR products from negative control (no reverse transcriptase) samples (*Figure 5B*). Sanger sequencing of the cloned PCR products confirmed transcription of *TP53RTG12* and *TP53RTG18/19* (*Figure 5—figure supplement 1*) in African elephant and *TP53RTG12* and *TP53RTG13* in Asian elephant fibroblast. We note that we used a Poly-T primer for cDNA synthesis, thus the amplification of *TP53RTG* transcripts indicates that these transcripts are poly-adenylated.

Most retrogenes lack native regulatory elements such as promoters and enhancers to initiate transcription, thus transcribed *TP53RTG* genes likely co-opted existing regulatory elements or evolved regulatory elements de novo. To identify putative transcriptional start sites and promoters of the highly expressed *TP53RTG12* gene we used geneid and GENESCAN to computationally predict exons in the African elephant gene and mapped the African and Asian elephant fibroblast RNA-Seq data onto scaffold_825, which encodes the *TP53RTG12* gene. We found that both computational methods predicted an exon ~2 kb upstream of the ENSEMBL annotated *TP53RTG12* gene, within an RTE-type non-LTR retrotransposon (RTE1_LA) that we annotated as Afrotherian-specific (*Figure 5C*). Consistent with this region encoding a transcribed 5'-UTR, a peak of reads mapped within the predicted 5'-exon and within the RTE1_LA retrotransposon (*Figure 5C*).

We attempted to identify the transcription start site of the *TP53RTG12* gene using several 5'-RACE methods, however, we were unsuccessful in generating PCR products from either African Elephant fibroblast or placenta cDNA, or Asian elephant fibroblast cDNA. Therefore, we designed a set of 34 PCR primers tiled across the region of scaffold_825 that encodes the *TP53RTG12* gene and used these primers to amplify PCR products from African and Asian Elephant fibroblast cDNA generated from DNase treated RNA. We then reconstructed the likely *TP53RTG12* promoter, transcription start site, and exon-intron structure from the pattern of positive PCR products. These data suggest that the major transcription initiation site of *TP53RTG12* is located within a RTE1_LA class transposable element (*Figure 5C*).

Next we tested the ability of the African and Asian elephant RTE1_LA elements and the RTE1_LA consensus sequence (as a proxy for the ancestral RTE1_LA sequence) to function as a promoter in transiently transfected African and Asian elephant fibroblasts when cloned into the promoterless pGL4.10[*luc2*] luciferase reporter vector. We found that the African and Asian elephant RTE1_LA elements increased luciferase expression 3.03-fold (t-test, p=2.41 $\times$ 10$^{-8}$) and 1.37-fold (t-test, p=2.60 $\times$ 10$^{-4}$), respectively, compared to empty vector controls (*Figure 5D*). However, luciferase expression from the pGL4.10[*luc2*] vector containing the RTE1_LA sequence was not significantly different than the empty vector control in either Asian (0.96-fold; t-test, p=0.61) or African elephant fibroblasts (0.95-fold; t-test, p=0.37; *Figure 5D*). These data indicate that transcription of

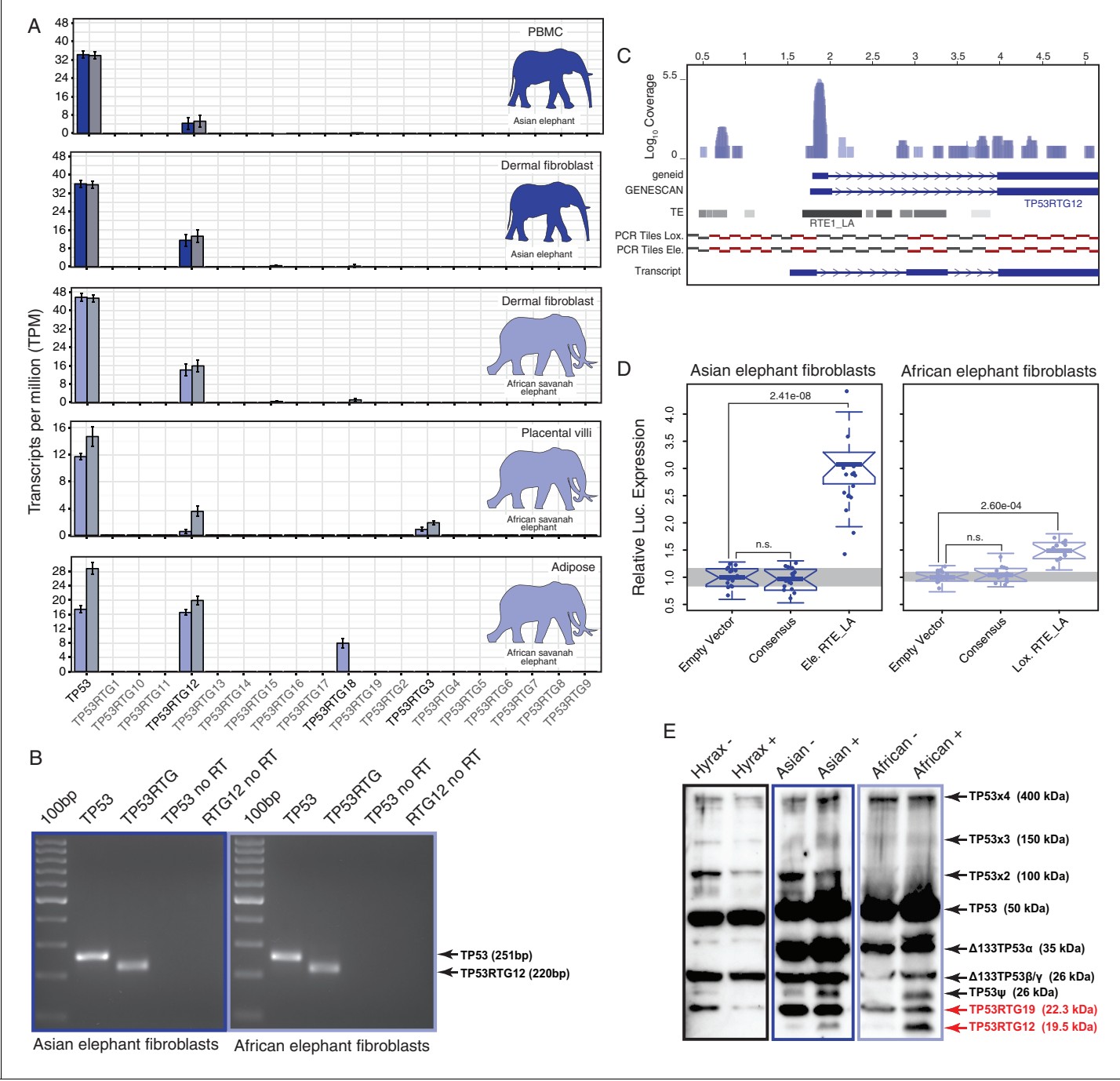

**Figure 5.** TP53RTG12 is transcribed and translated. (A) Transcription of elephant *TP53* and *TP53RTG* genes in dermal fibroblasts, white adipose, and placental villi. RNA-Seq data are shown as mean transcripts per million (TPM) with 95% confidence intervals of TPM value. Blue bars show TPM estimates from 'end-to-end' read mapping and gray bars shown 'local' read mapping. (B) qRT-PCR products generated Asian (left, blue sqaure) and African (right, light blue square) elephant fibroblast cDNA using primers specific to TP53 and TP53RTG12. cDNA was generated from DNaseI-treated RNA. No reverse transcriptase (no RT) controls for each qPCR reaction are shown, end point PCR products are shown. (C) Coverage of mapped reads from Asian (dark blue) and African (light blue) elephant fibroblast RNA-Seq data across the region of scaffold_885 encoding the *TP53RTG12* gene. The location of TP53RTG12 exons predicted from geneid and GENESCAN are shown in blue introns are shown as lines with arrows indicating the direction of transcription. Gray bars show the location of transposable elements around the *TP53RTG12* gene, darker gray indicates high sequence similarity to the consensus of each element. PCR tiles across this region are shown for African (Lox.) and Asian (Ele.) elephants, PCR primers generating amplicons are shown in red. The inferred *TP53RTG12* transcript is shown below. one kb scale shown from position one of African elephant (Broad/loxAfr3) scaffold_825. (D) Relative luciferase (Luc.) expression in Asian and African fibroblasts transfected with either the promoterless pGL4.10[*luc2*] luciferase

*Figure 5 continued on next page*

*Figure 5 continued*

reporter vector (empty vector), pGL4.10 containing the RTE_LA consensus sequences (Consensus), pGL4.10 containing the RTE_LA from Asian elephant (Ele. RTE_LA), or pGL4.10 containing the RTE_LA from African elephant (Lox. RTE_LA). Results are shown as fold difference in Luc. expression standardized to empty vector and *Renilla* controls. n = 16, Wilcoxon P-values. (E) Western blot of total cell protein isolated from South African Rock hyrax, Asian elephant (*Elephas*), and African elephant (*Loxodona*) dermal fibroblasts. −, control cells. +, cells treated with 50 J/m$^2$UV-C and the proteasome inhibitor MG-132. The name and predicted molecular weights of TP53 isoforms are shown.

The following figure supplements are available for figure 5:

**Figure supplement 1.** PCR and Sanger sequencing confirm *TP53RTG12* is transcribed in elephant fibroblasts.

**Figure supplement 2.** Unedited Western blots shown in *Figure 5E*.

TP53RTG12 likely initiates within a RTE1_LA-derived promoter, but that the ability of this RTE1_LA element to function as a promoter is not an ancestral feature of RTE1_LA elements.

## Elephant cells likely express TP53RTG proteins and numerous isoforms of TP53

To determine if *TP53RTG* transcripts are translated, we treated African elephant, Asian elephant, and hyrax cells with 50 J/m$^2$ UV-C (to stabilize TP53) and the proteasome inhibitor MG-132 (to block protein degradation), and assayed for TP53/TP53RTG proteins by Western blotting total cell protein with a polyclonal TP53 antibody (FL-393) that we demonstrated recognizes Myc-tagged TP53RTG12. We identified bands in both African and Asian elephant and hyrax total cell protein at the expected size for the full length p53, Δ133 p53β/γ, and p53ψ-like isoforms of the TP53 protein (*Khoury and Bourdon, 2010*) as well as high molecular weight bands corresponding to previously reported SDS denaturation resistant TP53 oligomers (*Cohen et al., 2008*; *Ottaggio et al., 2000*) and (poly)ubiquitinated TP53 conjugates (*Sparks et al., 2014*) (*Figure 5E*). We also identified an elephant-specific band at the expected size for the TP53RTG12 (19.6 kDa) and TP53RTG19 (22.3 kDa) proteins, suggesting that the *TP53RTG12* and *TP53RTG19* transcripts are translated in elephant fibroblasts (*Figure 5E*and *Figure 5—figure supplement 2*).

## Elephant cells have an enhanced TP53-dependent DNA-damage response

Our observation that *TP53RTG* genes are expressed suggests that elephant cells may have an altered TP53 signaling system compared to species without an expanded *TP53/TP53RTG* gene repertoire. To directly test this hypothesis we transiently transfected primary African elephant, Asian elephant, South African Rock hyrax (*Procavia capensis capensis*), East African aardvark (*Orycteropus afer lademanni*), and Southern Three-banded armadillo (*Tolypeutes matacus*) dermal fibroblasts with a luciferase reporter vector containing two TP53 response elements (pGL4.38[*luc2p*/p53 RE/Hygro]) and Renilla control vector (pGL4.74[*hRluc*/TK]). Next we used a dual luciferase reporter assay to measure the activation of the TP53 pathway in response to treatment with three DNA damage inducing agents (mitomycin C, doxorubicin, or UV-C) or nutlin-3a, which inhibits the interaction between MDM2 and TP53 and thus promotes TP53 signaling. We found that elephant cells generally up-regulated TP53 signaling in response to lower doses of each drug and UV-C than closely related species without an expanded TP53 gene repertoire (*Figure 6A*), indicating elephant cells have evolved an enhanced TP53 response.

To determine the consequences of an enhanced TP53 response we treated primary African and Asian elephant, hyrax, aardvark, and armadillo dermal fibroblasts with mitomycin C, doxorubicin, nutlin-3a, or UV-C and measured cell viability (live-cell protease activity), cytotoxicity (dead-cell protease activity), and the induction apoptosis (caspase-3/7 activation) using an ApoTox-Glo Triplex assay. Consistent with the results from the luciferase assay, we found that lower doses of mitomycin C or doxorubicin induced apoptosis in elephant cells than the other species (*Figure 6B*) and that the magnitude of the response was greater in elephant than other species (*Figure 6A*). Similarly UV-C exposure generally induced more elephant cells to undergo apoptosis than other species (*Figure 6A*). A striking exception to this trend was the response of elephant cells to the MDM2

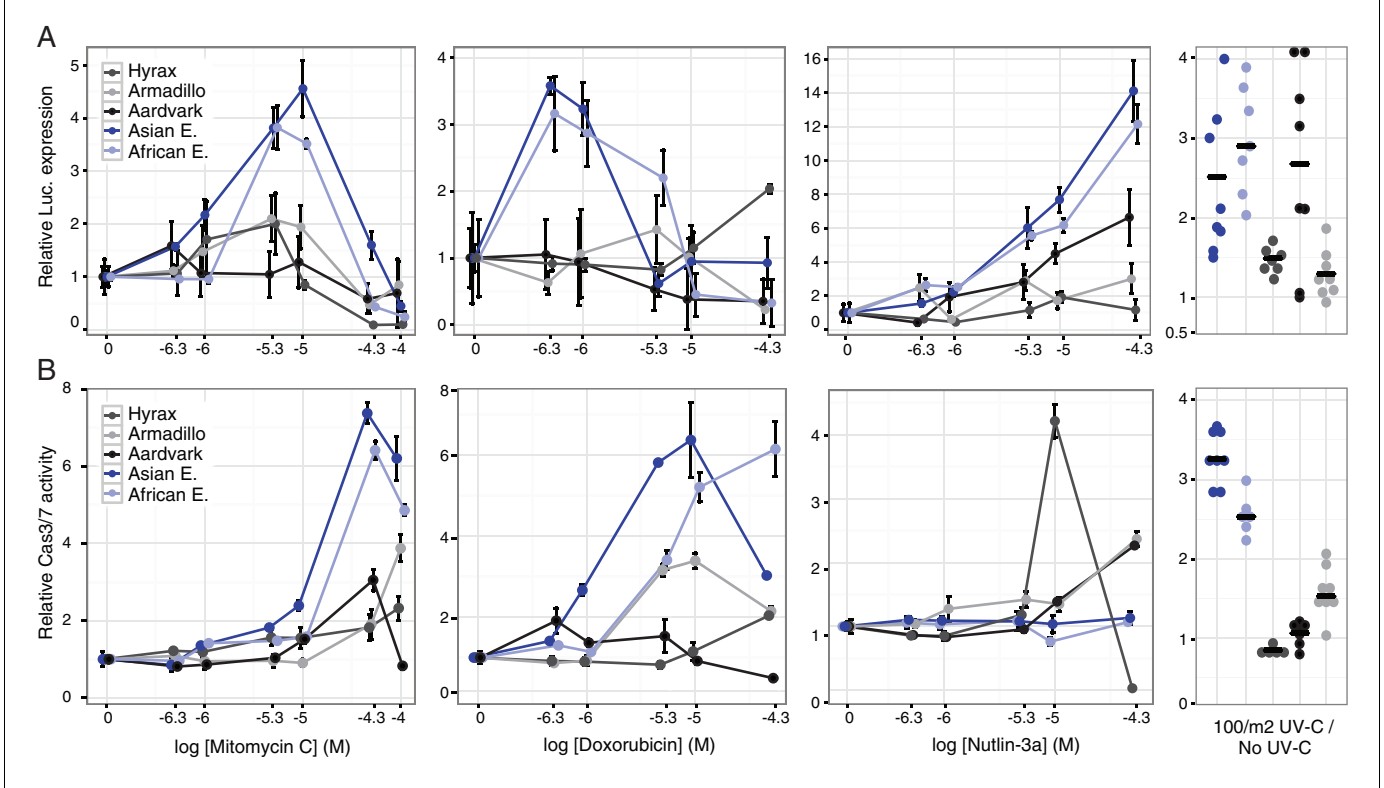

**Figure 6.** Elephant cells have enhanced TP53 signaling and are hyper-responsive to DNA damage. (**A**) Relative luciferase (Luc.) expression in African elephant, Asian elephant, hyrax, aardvark, and armadillo fibroblasts transfected with the pGL4.38[*luc2p*/p53 RE/Hygro] reporter vector and treated with either mitomycin c, doxorubicin, nutlin-3a, or UV-C. Data are shown as fold difference in Luc. expression 18 hr after treatment standardized to species paired empty vector and *Renilla* controls. n = 12, mean±SD. (**B**) Relative capsase-3/7 (Cas3/7) activity in African elephant, Asian elephant, hyrax, aardvark, and armadillo treated with either mitomycin c, doxorubicin, nutlin-3a, or UV-C. Data are shown as fold difference in Cas3/7 activity 18 hr after treatment standardized to species paired untreated controls. n = 12, mean±SD.

antagonist nutlin-3a, which elicited a strong TP53 transcriptional response (*Figure 6A*) but did not induce apoptosis (*Figure 6B*). Thus we conclude that elephant cells generally upregulate TP53 signaling and apoptosis at lower levels of DNA-damage than other species, but are resistant to nutlin 3 a induced apoptosis.

## TP53RTG genes are required for enhanced TP53 signaling and DNA-damage responses

To test if *TP53RTG* genes are necessary for the enhanced TP53-dependent DNA-damage response, we cotransfected African elephant fibroblasts with the pGL4.38[*luc2p*/p53 RE/Hygro] luciferase reporter vector, the pGL4.74[*hRluc*/TK] Renilla control vector, and either a TP53RTG-specific siRNA or a scrambled siRNA control (*Figure 7A*). Next we used a dual luciferase reporter assay to measure the activation of the TP53 pathway in response to treatment mitomycin C, doxorubicin, UV-C, or nutlin-3a. As expected given our previous results, we found that African elephant fibroblasts transfected with control siRNA induced TP53 signaling in response to each treatment (*Figure 7A*). In contrast, African elephant fibroblasts transfected with *TP53RTG*-specific siRNA had significantly lower luciferase expression, and thus reduced TP53 signaling, in response to either DNA-damaging agents (mitomycin C, doxorubicin, UV-C) or MDM2 antagonism (nutlin-3a). *TP53RTG* knockdown also elevated baseline TP53 signaling (*Figure 7B*). These data suggest that TP53RTG proteins have at least two distinct functions, inhibiting TP53 signaling in the absence of inductive signals and potentiation of TP53 signaling after the induction of DNA damage.

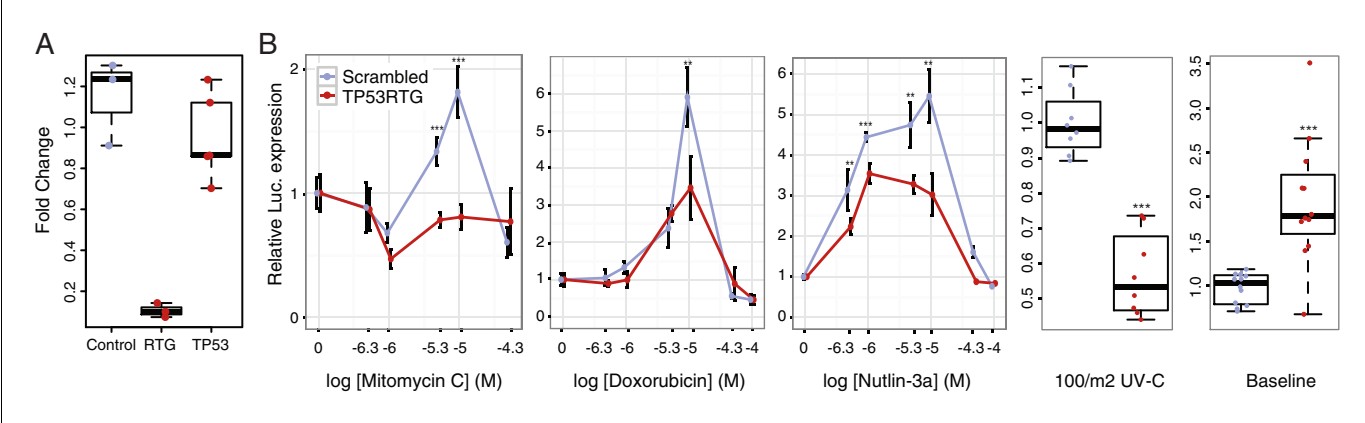

**Figure 7.** TP53RTG genes are required for enhanced TP53 signaling and DNA-damage responses. (**A**) Expression of TP53RTG and TP53 transcripts in African elephant fibroblasts treated with an siRNA to knockdown the expression of TP53RTG genes (red) or a scrambled (Control) siRNA (blue). Results are shown as fold-change in TP53RTG and TP53 transcript abundance relative to transcript abundance in scrambled siRNA control cells. The TP53RTG siRNA efficiently reduces the expression of TP53RTG transcripts, but does not reduce the expression of TP53 transcripts. (**B**) Relative luciferase (Luc.) expression in African elephant fibroblasts transfected with the pGL4.38[*luc2p*/p53 RE/Hygro] reporter vector and treated with an siRNA to knockdown the expression of TP53RTG genes (red) or a scrambled (negative control) siRNA (blue), and treated with either mitomycin c, doxorubicin, nutlin-3a, or UV-C. Data is shown as fold difference in Luc. expression 18 hr after treatment standardized to *Renilla* controls and no treatment. n > 4, mean±SD. **, Wilcoxon p>0.01. ***, Wilcoxon p>0.001.

## TP53RTG12 enhances TP53 signaling and DNA-damage responses via a transdominant mechanism

To test if TP53RTG12 is sufficient to mediate enhanced TP53 signaling and DNA-damage responses we synthesized the African elephant *TP53RTG12* gene (with mouse codon usage) and cloned it into the mammalian expression vector pcDNA3.1(+)/myc-His. We then transiently transfected mouse 3T3-L1 cells with the *TP53RTG12* pcDNA3.1(+)/myc-His expression vector and used the pGL4.38 [*luc2P*/p53 RE/Hygro] reporter system and ApoToxGlo assays to monitor activation of the TP53 signaling pathway and the induction of apoptosis in response to treatment with mitomycin C, doxorubicin, nutlin-3a, or UV-C. We found that heterologous expression of TP53RTG12 in mouse 3T3-L1 cells dramatically augmented luciferase expression from the pGL4.38[*luc2P*/p53 RE/Hygro] reporter vector in response to each treatment (*Figure 8A*) compared to empty vector controls, consistent with a enhancement of the endogenous TP53 signaling pathway. Similarly, expression of TP53RTG12 significantly augmented the induction of apoptosis in response to each treatment although the effect sizes were modest (*Figure 8B*). These data indicate that TP53RTG12 acts via a trans-dominant mechanism to enhance the induction of apoptosis by endogenous TP53 and that TP53RTG12 is sufficient to recapitulate at least some of the enhanced sensitivity of elephant cells to DNA damage. Furthermore, our observation that transfection with the *TP53RTG12* pcDNA3.1(+)/myc-His expression vector augments TP53 signaling and apoptosis suggests that the TP53RTG12 protein rather than transcript is responsible for these effects because the *TP53RTG12* transgene was synthetized with mouse codon usage and is only 73% (394/535 nts) identical to the elephant *TP53RTG12* gene.

TP53RTG proteins are unlikely to directly regulate TP53 target genes because they lack critical residues required for nuclear localization, tetramerization, and DNA-binding (*Figure 9A*). Previous studies, for example, have shown the TP53 mutants lacking the tetramerization domain and C-terminal tail are unable to bind DNA or transactivate luciferase expression from a reporter vector containing TP53 response elements (*Kim et al., 2012*). Similarly the p53Ψ isoform, which is truncated in the middle of the DNA binding domain and lacks the nuclear localization signal and oligomerization domain, is unable to bind DNA and is transcriptionally inactive (*Senturk et al., 2014*). Unexpectedly, we found that the GFP-tagged TP53RTG12 protein was both cytoplasmic and nuclear localized in transfected African elephant fibroblasts (*Figure 9B*), suggesting it interacts with another nuclear localized protein to enter the nucleus. Despite relatively strong nuclear localization, however, cells co-transfected with the *TP53RTG12* pcDNA3.1(+)/myc-His expression vector and the pGL4.38[*luc2P*/

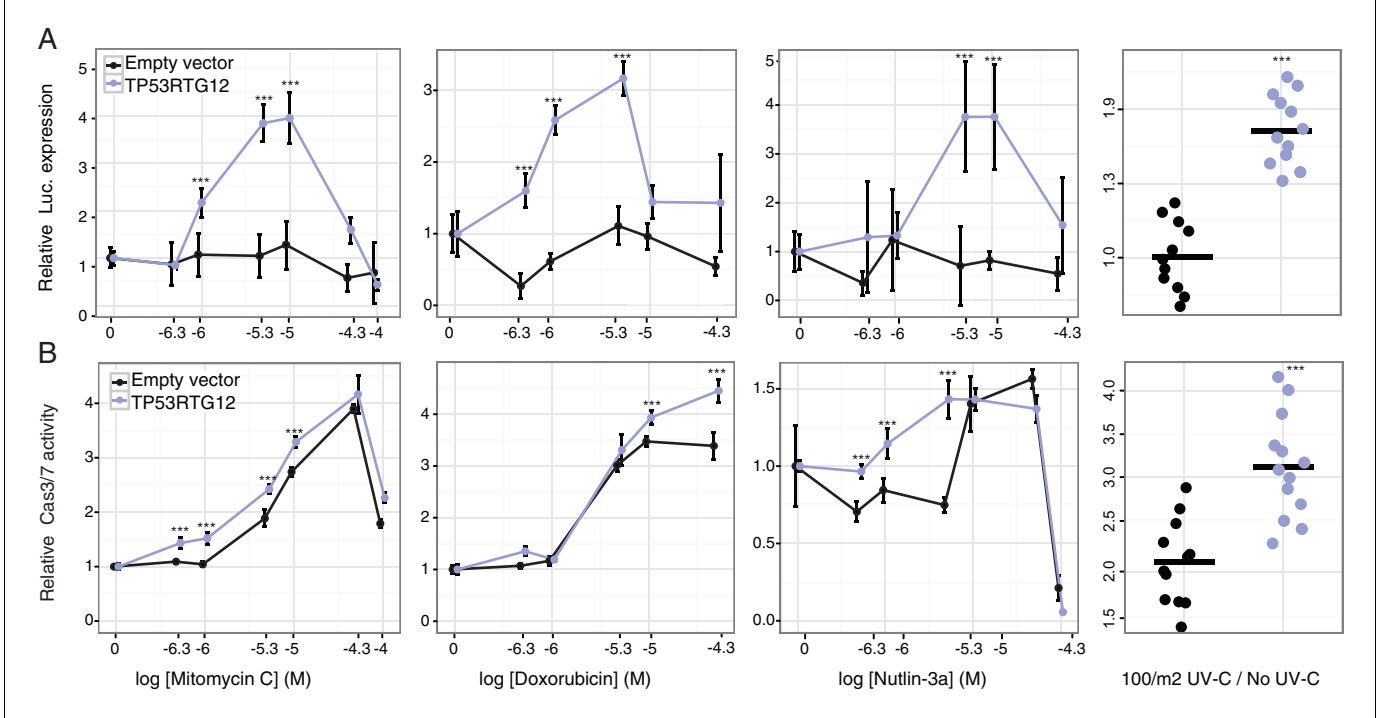

**Figure 8.** TP53RTG12 enhances TP53 signaling and DNA-damage responses. (**A**) Relative luciferase (Luc.) expression in mouse 3T3-L1 cells co-transfected with either the pGL4.38[*luc2P*/p53 RE/Hygro] Luc. reporter vector, *TP53RTG12* pcDNA3.1(+)/myc-His expression vector, or empty pcDNA3.1(+)/myc-His and treated with either mitomycin c, doxorubicin, nutlin-3a, or UV-C. Data is shown as fold difference in Luc. expression 18 hr after treatment standardized to cells transfected with only pGL4.38[luc2P/p53 RE/Hygro] and *Renilla* controls. n = 12, mean±SD. ***, Wilcoxon p>0.001. (**B**) Relative capsase-3/7 (Cas3/7) activity in mouse 3T3-L1 cells transfected with either the *TP53RTG12* pcDNA3.1(+)/myc-His expression vector or empty pcDNA3.1(+)/myc-His and treated with either mitomycin c, doxorubicin, nutlin-3a, or UV-C. Data is shown as fold difference in Cas3/7 activity 18 hr after treatment standardized to mock transfected. n = 12, mean±SD. ***, Wilcoxon p>0.001.

p53 RE/Hygro] luciferase reporter vector did not have elevated luciferase expression compared to controls, suggesting that TP53RTG12 is transcriptionally inactive or requires a cofactor to regulate target genes (*Figure 9C*).

## A model for TP53RTG function – decoy or guardian?

While TP53RTG12 does not appear to directly regulate gene expression, many of the TP53RTG proteins (including TP53RTG12) retain the MDM2 interaction motif in the transactivation domain and dimerization sites in the DNA binding domain (*Figure 9A*). These data suggest at least two non-exclusive models of TP53RTG action: (1) TP53RTG proteins may act as 'decoys' for the MDM2 complex allowing the canonical TP53 protein to escape negative regulation (*Figure 10A*); and (2) TP53RTG proteins may protect canonical TP53 from MDM2 mediated ubiquitination, which requires tetramerization (*Kubbutat et al., 1998*; *Maki, 1999*), by dimerizing with canonical TP53 and thereby preventing the formation of tetramers (*Figure 10F*).

The decoy model depends on the ability of TP53RTG proteins to physically interact with MDM2. Previous crystallographic studies of the TP53/MDM2 interaction have shown that a trio of residues in TP53 (F19, W23, and L26) insert deeply into a hydrophobic cleft in MDM2, which stabilizes the interaction (*Kussie et al., 1996*). We identified a W23G substitution in all TP53RTG proteins at a site that is invariant for tryptophan in TP53 proteins including African and Asian elephant TP53 (*Figure 10B*), suggesting that TP53RTG proteins may be unable to physically interact with MDM2. To infer the structural and functional consequences of the TP53RTG W23G substitution we generated a homology model of the elephant TP53RTG12/MDM2 complex using I-TASSER/ModRefiner (*Roy et al., 2010*; *Xu and Zhang, 2011*; *Zhang, 2008*) and the crystal structure of the MDM2/TP53 dimer as a template (*Kussie et al., 1996*). We found that the TP53RTG12 transactivation domain was inferred

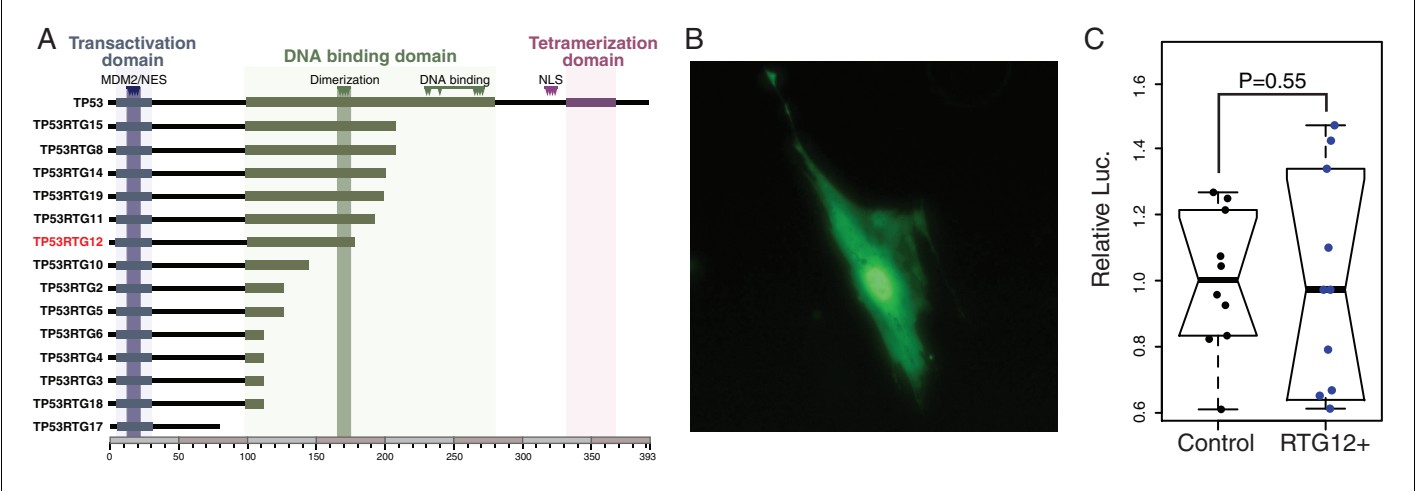

**Figure 9.** TP53RTG proteins are unlikely to directly regulate TP53 target genes. (**A**) Domain structure of TP53 and TP53RTG proteins. (**B**) Localization of TP53RTG12-GFP in African elephant fibroblasts. (**C**) Relative luciferase (Luc.) expression in African elephant fibroblasts transfected with co-transfected with either the *TP53RTG12* pcDNA3.1(+)/myc-His expression vector and the pGL4.38[*luc2P*/p53 RE/Hygro] luciferase reporter vector, or empty pcDNA3.1(+)/myc-His and the pGL4.38[*luc2P*/p53 RE/Hygro] luciferase reporter vector. Data are shown as fold difference in Luc. expression 48 hr after transfection standardized to *Renilla* and cells transfected with pcDNA3.1(+)/myc-His and pGL4.38[*luc2P*/p53 RE/Hygro]. n = 10.

to be a short α-helix (*Figure 10C*) and was very similar to the template structure (RMSD: 1.756), however, the W23G substitution is predicted to abolish crucial hydrophobic interactions between the amphipathic α-helix of TP53 and the hydrophobic cleft MDM2. Indeed, three methods (*Pires et al., 2014*) inferred that the W23G substitution is destabilizing on the TP53RTG12/MDM2 interaction (mCSM ΔΔG = −2.42, SDM ΔΔG = −5.46, DUET ΔΔG = −2.38) (*Figure 10D*). To experimentally test for an interaction between TP53RTG12 and MDM2 we transiently transfected HEK-293 cells with the TP53RTG12 pcDNA3.1(+)/myc-His expression vector, immunoprecipitated endogenous human MDM2, and assayed for co-immunoprecipitation of TP53RTG12 by Western blotting. While we efficiently co-immunoprecipitated endogenous human TP53 we did not co-immunoprecipitate Myc-tagged TP53RTG12 (*Figure 10E* and *Figure 10—figure supplement 1*), consistent with a lack of interaction between TP53RTG12 and MDM2.

Unlike the decoy model, the guardian model of TP53RTG function depends upon a physical interaction between TP53RTG and TP53. The TP53 dimer is stabilized by hydrophobic and polar interactions including a shell of nonpolar interactions formed by P177, H178, M243, and G244 and a stabilization network next to the nonpolar layer formed by charged residues from the two monomers (R181, E180, and R174). In addition, several polar and charged residues nearby but not within the dimerization interface contribute to the stability of the interacting monomers including D184 with R175, most of which are conserved in TP53RTG proteins (*Figure 10G*). To infer if derived residues in the TP53RTG12 dimerization interface might disrupt a physical interaction between TP53RTG12 and TP53 we generated a homology model of the TP53RTG12/TP53 dimer using I-TASSER/ModRefiner (*Roy et al., 2010*; *Xu and Zhang, 2011*; *Zhang, 2008*) and the crystal structure of the TP53 tetramer as a template (*Kitayner et al., 2006*). We found that the TP53RTG12 dimerization interface was inferred to be a 2–4 residue α-helix (*Figure 10H*) that was nearly identical to the template structure (RMSD: 0.605), suggesting TP53RTG12-specific residues in dimerization interface are unlikely to disrupt the structure of the interface. Unlike the MDM2 interaction site, TP53RTG12-specific substitutions were predicted to maintain intermolecular hydrophobic interactions with TP53 (*Figure 10H*). Consistent with maintenance of dimerization potential, the net ΔΔG of the derived amino acid substitutions in the TP53RTG12 dimerization interface were under 2 (*Figure 10I*). To experimentally test for an interaction between TP53RTG12 and TP53 we transiently transfected HEK-293 cells with the TP53RTG12 pcDNA3.1(+)/myc-His expression vector, immunoprecipitated TP53RTG12 with a Myc antibody, and assayed for co-immunoprecipitation of endogenous human TP53 by Western blotting. We found that Myc-tagged TP53RTG12 efficiently co-immunoprecipitated endogenous human TP53

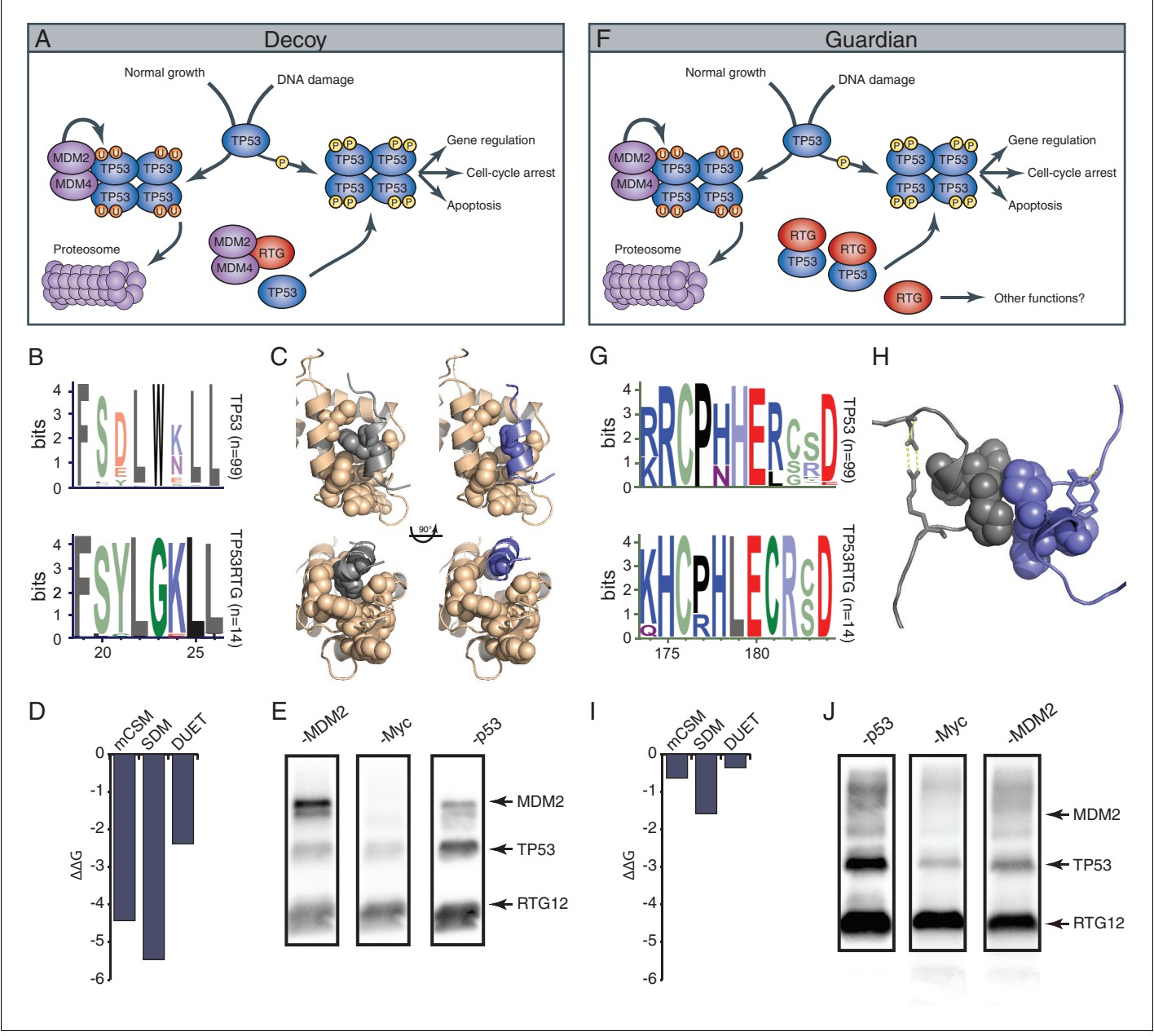

**Figure 10.** TP53RTG12 interacts with TP53 but not MDM2. (A) Decoy model of TP53RTG12 function. Under normal conditions TP53 is negatively regulated by the MDM complex which ubiquitinates TP53 tetramers leading to proteosomal degradation. Upon DNA damage TP53 is phosphorylated, preventing interaction with the MDM complex and activating downstream TP53 signaling. In elephant cells TP53RTG12 may dimerize with the MDM2 complex, allowing TP53 to escape negative regulation. (B) Logo of the MDM2 interaction motif from 99 mammalian TP53 proteins (upper) and TP53RTG proteins (lower). (C) Structure model of the MDM2/TP53 (left) and MDM2/TP53RTG12 (right) interaction. MDM2 is shown in tan with hydrophobic residues that mediate the interaction with TP53 as spheres, TP53 in gray with W23 as a sphere, and TP53RTG12 shown in blue with G23 as a sphere. (D) Predicted effects of the W23G substitution (ΔΔG) on the stability of the MDM2/TP53RTG12 interaction estimated with mCSM, SDM, and DUET. (E) HEK-293 cells were transiently transfected with the TP53RTG12 pcDNA3.1(+)/myc-His expression vector and total cell protein immunoprecipitated with an α-MDM2 antibody. Co-immunoprecipitation of Myc-tagged TP53RTG12 and TP53 were assayed by Western blotting with α-Myc, α-TP53, and α-MDM2 antibodies after chemically stripping the blot, respectively. Although TP53 was co-immunoprecipitated, Myc-tagged TP53RTG12 was not. (F) Rather than interfering with the interaction between the MDM complex and TP53 as in the 'Decoy' model, TP53RTG12 may dimerize with canonical TP53 and block formation of TP53 tetramers. These TP53RTG12/TP53 dimers cannot be ubiquitinated generating a pool of TP53 proteins to rapidly respond to lower levels or DNA damage and stress than other species. (G) Logo of the dimerization domain of TP53 from 99 mammals (upper) and TP53RTG proteins (lower). (H) Model of the TP53/TP53RTG12 interaction. Residues critical for dimerization are shown as spheres, note sites involved in dimerization are conserved in TP53RTG proteins. (I) Predicted effects of the W23G substitution (ΔΔG) on the stability of the

*Figure 10 continued on next page*

Figure 10 continued

MDM2/TP53RTG12 interaction estimated with mCSM, SDM, and DUET. (J) HEK-293 cells were transiently transfected with the TP53RTG12 pcDNA3.1 (+)/myc-His expression vector and total cell protein immunoprecipitated with an α-Myc antibody. Co-immunoprecipitation of TP53 and MDM2 were assayed by serial Western blotting with α-Myc, α-TP53, and α-MDM2 antibodies after chemically stripping the blot, respectively.

The following figure supplement is available for figure 10:

**Figure supplement 1.** Uncropped Western blots shown in *Figure 8B*.

(*Figure 10J*). These data are consistent with a physical interaction between TP53RTG12 and TP53 but not between TP53RTG12 and MDM2, supporting the guardian model.

## Discussion

A major developmental and life history constraint on the evolution of large body sizes and long life-spans in animals is an increased risk of developing cancer. There is no correlation, however, between body size or lifespan and cancer risk because large and long-lived organisms have evolved enhanced cancer suppression mechanisms that delay the development of cancer until post-reproduction (when selection generally cannot act). This simple evolutionary rational demands mechanistic explanations (*Peto, 2015*), which have thus far been elusive. Here we show that the master tumor suppressor TP53, which is essential for preventing cancer because it triggers proliferative arrest and apoptosis in response to a variety of stresses such as DNA damage, was retroduplicated in the Paenungulate stem-lineage and rapidly increased in copy number through repeated segmental duplications during with the evolution of Proboscideans. The expansion of the *TP53RTG* gene family occurred coincident with the evolution of large body sizes and enhanced sensitivity of elephant cells to genotoxic stress, suggesting that Proboscideans resolved Peto's paradox at least in part through the evolution of aug-mented TP53 signaling.

### Comparison to previous studies of elephant TP53

Previous studies have suggested that the *TP53* gene family expanded in the elephant lineage (*Abegglen et al., 2015*; *Caulin and Maley, 2011*; *Caulin et al., 2015*), however, these studies did not establish the mechanism by which the *TP52RTG* gene family expanded. Several potential mecha-nisms could have increased *TP52RTG* copy number, including serial (independent) retrotranspostion from the parent *TP53* gene, a single retrotranspostion event followed by repeated rounds of seg-mental duplication of *TP52RTG* containing loci, retrotransposition of expressed *TP52RTG* genes, or some combination of these models. Each model is associated with a distinct set of genomic 'finger-prints'. If copy number expanded through independent retrotransposition events, for example, *TP53RTG* encoding regions of the genome will not be homologous whereas the model of a single retrotranspostion event followed by repeated rounds of segmental duplication predicts that the TP53RTG encoding loci will be homolgous. Consistent with copy number expansion through a single retrotransposition event followed by repeated rounds of segmental duplication, we found that flak-ing regions of each *TP52RTG* locus were homologous and contained the same unique combination of transposable elements. Indeed, the 3'-end of each duplicate terminates at a ~5 kb long L1MB5 LINE element suggesting that transposable element mediated recombination may have played a role in promoting segmental duplication.

If *TP53RTG* copy number expansion played a causal role in evolution of enhanced cancer resis-tance in elephants then the gene family must have expanded prior to or coincident with the evolu-tion of increased body sizes in Proboscideans rather than after the evolution of large bodies but before the African and Asian elephant lineages diverged ~8 MYA (*Rohland et al., 2010*, *2007*). Thus dating the expansion of the *TP53RTG* gene family is essential for determining if *TP53RTG* genes played a role in the evolution of enhanced cancer resistance in elephants. Previous studies, however, did not establish when *TP53RTG* copy number expanded in the evolution of Proboscideans. Fortu-nately our observation that copy number expansion occurred through segmental duplications allowed us to use molecular phylogenetic methods to date each duplication event. These data indi-cate that the initial *TP53RTG* retrotransposition event occurred in the Paenungulate stem-

lineage ~ 64 MYA, followed the rapid expansion after ~40 MYA. We also found that the increase in *TP53RTG* copy number occurred coincident with the evolution of large bodies in the Proboscidean lineage, implicating copy number expansion in the resolution of Peto's paradox.

While *Abegglen et al. (2015)* found that elephant and human cells had different sensitivities to ionizing radiation, their taxon sampling did not allow for polarizing which species was different. Indeed, their taxon sampling does not allow for a Eutherian out-group. To determine if elephants have a derived sensitivity to genotoxic stress, we compared the response of fibroblasts from an African savannah elephant, an Asian elephant, and closely related out-group species to DNA damage inducing agents. Our out-group species included the South African Rock hyrax, the closest living relative of elephants, the East African aardvark, an Afrotherian from the sister lineage of the Paenungulates, and the Southern Three-banded armadillo, an Atlantogenatan from the sister lineage of the Afrotherians. Our results indicate that elephant cells are particularly sensitive to genotoxic tress, suggesting that this sensitivity evolved coincident with the evolution of large body sizes and an expanded *TP53* gene repertoire in Proboscideans.

## An embarrassment of riches?

Our observation that the elephant genome contains 19 *TP53RTG* genes raises numerous questions: Do these loci, for example, encode functional genes or pseudogenes and what processes underlie copy number expansion? Answers to these questions are essential for understanding whether *TP53RTG* genes are casually related to the resolution of Peto's paradox in Proboscideans or if they are irrelevant relicts of ancient transposition events, like so many other pseudogenes that riddle mammalian genomes. Unfortunately, it is difficult to answer these questions.

Which *TP53RTG* loci encode functional genes and which encode pseudogenes is not easy to infer. Many duplicate genes are preserved because they evolve tissue-specific or developmental-stage specific expression patterns (subfunctionalization), new functions (neofunctionalization) that resolve redundancy between duplicates, or reduced expression levels that preserves correct expression dosage. Exhaustively characterizing gene expression in elephant tissues is difficult because appropriate tissue samples are unavailable, therefore we are unable to definitively determine which *TP53RTG* loci are transcribed. Our RNA-Seq and RT-PCR/Sanger sequencing data indicate that at least *TP53RTG12*, *TP53RTG18/19*, and *TP53RTG13* are transcribed in dermal fibroblasts. *Abegglen et al. (2015)* used RT-PCR and Sanger sequencing to show two distinct transcripts were expressed in elephant PBMCs, but they did not assign the loci to which these transcripts correspond. We analyzed the chromatograms shown in Abegglen et al. Figure 4 and found that the 185 bp product is likely a transcript from the *TP553RTG14* gene and the 201 bp product is likely a transcript from the *TP553RTG5* gene. Thus, our combined data suggest that at least five *TP53RTG* genes are transcribed. Furthermore we did not observe *TP553RTG14* or *TP553RTG5* expression in adipose, placenta, or fibroblasts suggesting that the expression of some *TP53RTG* genes is tissue-specific.

The large number of *TP53RTG* loci in elephants combined with the observation that only five are transcribed suggests that this gene family may evolve by a birth and death process, in which new genes are created by duplication and some duplicates are maintained in the genome whereas others become nonfunctional or deleted similar to other large genes families such as histones (*González-Romero et al., 2010*; *Rooney et al., 2002*) and venom genes (*Lynch, 2007*). Under this model selection acts to maintain a minimal number of functional copies (functional copy number) rather than total copy number, the increase in total copy number is driven by the total number of loci and the rates of duplication, loss, and fixation. Thus the overall increase in total *TP53RTG* copy number may be a selectively neutral process, driven simply by higher rates of duplication and/or fixation than loss.

## Is TP53RTG12 a decoy, a guardian, or something else?

Our results demonstrate that elephant cells induce TP53 signaling and trigger apoptosis at lower thresholds of genotoxic stress than closely related species without an expanded *TP53* repertoire and that this reduced sensitivity is dependent upon *TP53RTG* genes. Furthermore, heterologous expression of TP53RTG12 in mouse cells was sufficient to augment endogenous TP53 signaling and recapitulate an elephant-like sensitivity to genotoxic stress, indicating that TP53RTGs acts through a transdominant mechanism. While the mechanism of action is unclear, *TP53RTG* genes may augment

TP53 signaling through several non-exclusive mechanisms including functioning as non-coding RNAs (*Poliseno et al., 2010*), protein 'decoys' for the MDM2 complex that allowing canonical TP53 to escape negative regulation (*Abegglen et al., 2015*), and protein 'guardians' that protect canonical TP53 from MDM2 mediated ubiquitination.

Consistent with the guardian model, we found that Myc-tagged TP53RTG12 efficiently co-immu-noprecipitated with endogenous TP53 but not with endogenous MDM2. The lack of an interaction between TP53RTG12 and MDM2 likely results from a W23G substitution that is predicted to abolish a crucial hydrophobic interaction between the amphipathic α-helix of TP53 and the hydrophobic cleft MDM2. The W23G substitution is found in all TP53RTG proteins but not African and Asian elephant TP53, indicating that it occurred before the expansion of the *TP53RTG* gene family and likely prevents interaction of any TP53RTG protein and MDM2. Previous studies have shown that tetramerization of TP53 is required for its efficient MDM2-mediated ubiquitination (*Kubbutat et al., 1998*; *Maki, 1999*), suggesting that TP53RTG proteins may dimerize with and protect TP53 from ubiquitination thereby contributing to a standing pool of TP53 that is able to rapidly respond to DNA damage. Consistent with this mechanism, transgenic mice with an increase in TP53 copy number (*García-Cao et al., 2002*) or a hypomorphic *Mdm2* allele have elevated basal TP53 activity and are resistant to tumor formation (*Mendrysa et al., 2006*), indicating that shifting the TP53-MDM2 equilibrium away from TP53 degradation can directly promote cancer resistance.

The 'decoy' model (*Abegglen et al., 2015*) has also been challenged because it would allow for activation of the TP53 signaling pathway in the absence of DNA damage (*Perez and Komiya, 2016*), which is generally lethal in animal models (*Hoever et al., 1994*; *Lozano, 2010*). Thus if TP53RTG proteins allow TP53 to escape negative regulation by MDM2, how do elephants tolerate elevated basal TP53 levels? Clearly further studies are required to specifically test whether TP53RTG proteins interfere with the interaction between the MDM2 complex and TP53, protect TP53 from ubiquitination, or have other functions.

## What do extra *TP53* genes cost?

Our observation reveals that functional *TP53* duplicates only occur in the elephant lineage (and perhaps some bats) suggests that increased *TP53* dosage has a cost. Previous studies found that transgenic mice that overexpress *Trp53* were cancer resistant but had major life history tradeoffs including slower pre- and post-natal growth rates and reduced size (*Maier et al., 2004*), a shortened lifespan (*Maier et al., 2004*), accelerated aging (*Tyner et al., 2002*), and reduced fertility (*Allemand et al., 1999*; *Maier et al., 2004*), as well as developmental tradeoffs including reduced proliferation, cellularity, and atrophy across multiple organ and tissue systems (*Dumble et al., 2007*; *Maier et al., 2004*), defective ureteric bud differentiation, and small kidneys (*Godley et al., 1996*). Thus, increases in TP53 copy number protects against cancer but appears to come with the cost of developmental delays, accelerated aging, and reduced fertility (*Campisi, 2003*; *Donehower, 2002*; *Ferbeyre and Lowe, 2002*; *Rodier et al., 2007*).

Reduced male fertility appears to be a particularly expensive cost of increased *TP53* dosage. *Allemand et al., 1999*), for example, generated transgenic lines of mice with one (MTp53-176), two (MTp53-112), or 15 (MTp53-94) extra copies of the Trp53 gene fused to the inducible promoter of the metallothionein I (MT) gene. They found that transgenic males with the highest Trp53 dosage were nearly infertile because the majority of developing spermatids underwent apoptosis before developing into mature sperm, males with intermediate dosage were subfertile and produced sperm with abnormal morphologies (teratozoospermia) indicative of defective terminal differentiation of postmeiotic cells, whereas males with the lowest dosage were fertile. Similarly, *Maier et al. (2004)* generated a transgenic line that overexpresses the p44 isoform of *Trp53* (Δ40 p53) and found that both males and females exhibited shortened reproductive lifespans. Males, however, were more severely affected than females with a catastrophic loss of sperm-producing cells and massive degeneration of the seminiferous epithelium leading to a 'Sertoli-cell only' phenotype.

In contrast to transgenic mice that either overexpress full length TP53 or the Δ40 p53 isoform, transgenic 'super p53' mice with one (p53-tg) or two (p53-tg[b]) additional copies of the endogenous TP53 locus have an enhanced DNA-damage response and are tumor resistant, yet age normally and are fertile (*García-Cao et al., 2002*, *2006*; *Matheu et al., 2007*). Enhanced tumor suppression and normal aging is also observed in *Mdm2^{puro/Δ7−12}* transgenic mice, which have one hypomorphic and one null allele of *Mdm2*, express ~30% of the wild-type level of Mdm2, and have constitutively high

TP53 activity (*Mendrysa et al., 2006*, *2003*). These results suggest that the costs of increased *TP53* copy number are incurred above a threshold of about 2–3 extra copies and can be reduced by maintaining normal regulation of *TP53* transcription and negative post-transcriptional regulation by MDM2. Consistent with this model, deletion of either *Mdm2* or *Mdm4* rescues *Trp53*-dependent embryonic lethality in mice (*Finch et al., 2002*; *Migliorini et al., 2002*; *Parant et al., 2001*) and zebrafish (*Chua et al., 2015*).

Collectively, these data indicate that increased *TP53* dosage comes with a cost, and suggest that Proboscideans evolved a mechanism that reduced this cost, broke a major developmental and evolutionary constraint on *TP53* copy number, or perhaps evaded paying the cost of increased *TP53* copy number altogether. *TP53RTG* genes, for example, are transcribed from a transposable element derived promoter that is evolutionarily younger than the retrogenes. Thus, the initial TP53RTG genes were unlikely to be transcribed and incur a cost. Similarly, our phylogenetic analyses indicates that copy number expanded after the initial *TP53RTG* gene acquired several mutations, including a premature stop codon that terminates the protein before the DNA-binding domain, which likely reduces or completely eliminates the costs associated with TP53 target gene regulation. Although more detailed evolutionary and comparative analyses are required to determine if *TP53RTG* genes incurred a cost, it is possible that the costs were minimized because functional *TP53RTG* genes evolved through non-functional intermediates, which accumulated loss of function mutations that minimized redundancy with *TP53*.

## Materials and methods

### Identification of *TP53/PT53RTG* genes in sarcopterygian genomes

We used BLAT to search for *TP53* genes in 61 Sarcopterygian genomes using the human TP53 protein sequences as an initial query. After identifying the canonical *TP53* gene from each species, we used the nucleotide sequences corresponding to this *TP53* CDS as the query sequence for additional BLAT searches within that species genome. To further confirm the orthology of each *TP53* gene we used a reciprocal best BLAT approach, sequentially using the putative CDS of each *TP53* gene as a query against the human genome; in each case the query gene was identified as TP53. Finally, we used the putative amino acid sequence of the TP53 protein as a query sequence in a BLAT search.

We thus used BLAT to characterize the *TP53* copy number in Human (*Homo sapiens*; GRCh37/hg19), Chimp (*Pan troglodytes*; CSAC 2.1.4/panTro4), Gorilla (*Gorilla gorilla gorilla*; gorGor3.1/gorGor3), Orangutan (*Pongo pygmaeus abelii*; WUGSC 2.0.2/ponAbe2), Gibbon (*Nomascus leucogenys*; GGSC Nleu3.0/nomLeu3), Rhesus (*Macaca mulatta*; BGI CR_1.0/rheMac3), Baboon (*Papio hamadryas*; Baylor Pham_1.0/papHam1), Marmoset (*Callithrix jacchus*; WUGSC 3.2/calJac3), Squirrel monkey (*Saimiri boliviensis*; Broad/saiBol1), Tarsier (*Tarsius syrichta*; Tarsius_syrichta2.0.1/tarSyr2), Bushbaby (*Otolemur garnettii*; Broad/otoGar3), Mouse lemur (*Microcebus murinus*; Broad/micMur1), Chinese tree shrew (*Tupaia chinensis*; TupChi_1.0/tupChi1), Squirrel (*Spermophilus tridecemlineatus*; Broad/speTri2), Mouse (*Mus musculus*; GRCm38/mm10), Rat (*Rattus norvegicus*; RGSC 5.0/rn5), Naked mole-rat (*Heterocephalus glaber*; Broad HetGla_female_1.0/hetGla2), Guinea pig (*Cavia porcellus*; Broad/cavPor3), Rabbit (*Oryctolagus cuniculus*; Broad/oryCun2), Pika (*Ochotona princeps*; OchPri3.0/ochPri3), Kangaroo rat (*Dipodomys ordii*; Broad/dipOrd1), Chinese hamster (*Cricetulus griseus*; C_griseus_v1.0/criGri1), Pig (*Sus scrofa*; SGSC Sscrofa10.2/susScr3), Alpaca (*Vicugna pacos*; Vicugna_pacos-2.0.1/vicPac2), Dolphin (*Tursiops truncatus*; Baylor Ttru_1.4/turTru2), Cow (*Bos taurus*; Baylor Btau_4.6.1/bosTau7), Sheep (*Ovis aries*; ISGC Oar_v3.1/oviAri3), Horse (*Equus caballus*; Broad/equCab2), White rhinoceros (*Ceratotherium simum*; CerSimSim1.0/cerSim1), Cat (*Felis catus*; ICGSC Felis_catus 6.2/felCat5), Dog (*Canis lupus familiaris*; Broad CanFam3.1/canFam3), Ferret (*Mustela putorius furo*; MusPutFur1.0/musFur1), Panda (*Ailuropoda melanoleuca*; BGI-Shenzhen 1.0/ailMel1), Megabat (*Pteropus vampyrus*; Broad/pteVam1), Microbat (*Myotis lucifugus*; Broad Institute Myoluc2.0/myoLuc2), Hedgehog (*Erinaceus europaeus*; EriEur2.0/eriEur2), Shrew (*Sorex araneus*; Broad/sorAra2), Minke whale (*Balaenoptera acutorostrata scammoni*; balAcu1), Bowhead Whale (*Balaena mysticetus*; v1.0), Rock hyrax (*Procavia capensis*; Broad/proCap1), Sloth (*Choloepus hoffmanni*; Broad/choHof1), Elephant (*Loxodonta africana*; Broad/loxAfr3), Cape elephant shrew (*Elephantulus edwardii*; EleEdw1.0/eleEdw1), Manatee (*Trichechus manatus latirostris*; Broad v1.0/triMan1), Tenrec (*Echinops telfairi*; Broad/echTel2), Aardvark (*Orycteropus afer afer*; OryAfe1.0/

oryAfe1), Armadillo (*Dasypus novemcinctus*; Baylor/dasNov3), Opossum (*Monodelphis domestica*; Broad/monDom5), Tasmanian devil (*Sarcophilus harrisii*; WTSI Devil_ref v7.0/sarHar1), Wallaby (*Macropus eugenii*; TWGS Meug_1.1/macEug2), Platypus (*Ornithorhynchus anatinus*; WUGSC 5.0.1/ornAna1), Medium ground finch (*Geospiza fortis*; GeoFor_1.0/geoFor1), Zebra finch (*Taeniopygia guttata*; WashU taeGut324/taeGut2), Budgerigar (*Melopsittacus undulatus*; WUSTL v6.3/melUnd1), Chicken (*Gallus gallus*; ICGSC Gallus_gallus-4.0/galGal4), Turkey (*Meleagris gallopavo*; TGC Turkey_2.01/melGal1), American alligator (*Alligator mississippiensis*; allMis0.2/allMis1), Painted turtle (*Chrysemys picta bellii*; v3.0.1/chrPic1), Lizard (*Anolis carolinensis*; Broad AnoCar2.0/anoCar2), X. tropicalis (*Xenopus tropicalis*; JGI 7.0/xenTro7), Coelacanth (*Latimeria chalumnae*; Broad/latCha1).

### *TP53/TP53RTG* copy number estimation in proboscidean genomes

### Identification of TP53/TP53RTG genes in the Asian elephant genome

We used previously published whole genome shotgun sequencing data from an Asian elephant (*Elephas maximus*) generated on an Illumina HiSeq 2000{'Hou'} to estimate the *TP53/TP53RTG* copy number in the genome. For these analyses, we combined data from two individual elephants (ERX334765 and ERX334764) into a single dataset of 151,482,390 76-nt paired end reads. We then mapped Asian elephant reads onto the African elephant *TP53/TP53RTG* genes using Bowtie2 in paired-end mode, with the local alignment and 'very sensitive' options and Cufflinks (version 0.0.7) to assemble mapped reads into genes which we treated as putative Asian elephant orthologs. We identified 14 putative 1:1 orthologous TP53RTG genes in the Asian elephant genome, including *TP53RTG1, TP53RTG2, TP53RTG3, TP53RTG4, TP53RTG5, TP53RTG6, TP53RTG9, TP53RTG10, TP53RTG11, TP53RTG13, TP53RTG14, TP53RTG17, TP53RTG18, and TP53RTG19*.

### Identification of TP53/TP53RTG genes in the Columbian and woolly mammoth genomes

We used previously published whole genome shotgun sequencing data from an ~11,000 year old Columbian mammoth (*Mammuthus columbi*) generated on an Illumina HiSeq 1000{'Sparks'} to estimate the *TP53/TP53RTG* copy number in the genome. For these analyses, we combined data from three individual mammoths (SRX329134, SRX329135, SRX327587, SRX327586, SRX327583, SRX327582) into a single dataset of 158,704,819 102-nt paired end reads. We then mapped Asian elephant reads onto the African elephant *TP53/TP53RTG* genes using Bowtie2 in paired-end mode, with the local alignment and 'very sensitive' options, and Cufflinks (version 0.0.7) to assemble mapped reads into genes which we treated as putative Asian elephant orthologs. We identified 14 putative 1:1 orthologous *TP53RTG* genes in the Columbian mammoth genome, including *TP53RTG10, TP53RTG11, TP53RTG13, TP53RTG14, TP53RTG16, TP53RTG17, TP53RTG19, TP53RTG2, TP53RTG3, TP53RTG4, TP53RTG5, TP53RTG6, TP53RTG8, TP53RTG9*.

### Identification of TP53/TP53RTG genes in the American mastodon genome

We used previously published whole genome shotgun sequencing data generated on a Roche 454 Genome Sequencer (GS FLX) to estimate the TP53 copy number in a 50,000–130,000 year old American mastodon (Mammut americanum){'Langmead'}. To estimate TP53 copy number we combined 454 sequencing data from 'Library A' (SRX015822) and 'Library B' (SRX015823) into a single dataset of 518,925 reads. Reads were converted from FASTQ to FASTA and aligned to African elephant *TP53* gene and retrogene sequences using Lastz. We used the 'Roche-454 90% identity' mapping mode, not reporting matches lower than 90% identity. Three American mastodon reads were identified that mapped to at least one TP53 gene/retrogene. The likely identities of these reads were determined by using BLAT to map their location in the African elephant (Broad/loxAfr3) genome; mastodon reads mapping to a single elephant gene with >98% identity were considered likely mastodon orthologs. This strategy mapped one read to the *TP53TRG8* retrogene, one read to the either the *TP53TRG3* or *TP53RTG10* retrogenes, and one read to the *TP53TRG6* retrogene.

### Gene tree reconciliation

For gene tree reconciliation putative Asian elephant, Columbian mammoth, and woolly mammoth *TP53/TP53RTG* orthologs were manually placed into the African elephant *TP53/TP53RTG* gene tree. We then used Notung v2.6 (*Chen et al., 2000*) to reconcile the gene and species trees, and

interpreted Asian elephant-, Columbian mammoth-, and wooly mammoth-specific gene losses as unsampled *TP53RTG* genes if the putative divergence date of the ortholog predated the African-(Asian elephant/Mammoth) divergence. We thus inferred two unsampled genes (*TP53RTG7* and *TP53RTG8*) in the Asian elephant genome, and three unsampled genes (*TP53RTG18*, *TP53RTG7* and *TP53RTG1*) in the Columbian mammoth genome. Note that this method will estimate the minimum number of *TP53/TP53RTG* genes in the genome because our mapping strategy cannot identify lineage specific duplications after each species diverged from the African elephant lineage. Finally, we used Notung v2.6 (*Chen et al., 2000*) to reconcile the gene and species trees, and interpreted lineage-specific gene losses as unsampled *TP53RTG* genes (i.e., genes present in the genome but missing from the aDNA sequencing data). This process was modified slightly for American mastodon, such that 'lost' *TP53RTG* genes younger than the mastodon-elephant split were considered elephant-specific paralogs rather than unsampled genes present in the mastodon genome.

## Normalized read depth

We also used normalized mapped read depth to estimate *TP53/TP53RTG* copy number. Unlike gene tree reconciliation, copy number estimates based on read depth cannot resolve the orthology of specific *TP53/TP53RTG* copies but can estimate the total number of *TP53/TP53RTG* copies in the genome and thus provide evidence of lineage specific copy number changes. For this analyses, we mapped Asian elephant-, Columbian mammoth-, and woolly mammoth-specific reads onto the African elephant *TP53/TP53RTG* genes using Bowtie 2 (*Langmead and Salzberg, 2012*) with the local alignment and 'very sensitive' options and Cufflinks (version 0.0.7) (*Trapnell et al., 2012*) to assemble mapped reads into genes. We then summed the read depth ('coverage') of *TP53RTG* genes and normalized this summed *TP53RTG* read depth to the read depth of five single copy genomic regions equal in length to average *TP53RTG* gene length. This ratio represents an estimate of the *TP53/TP53RTG* copy number in the genome.

## Correlation between *TP53/TP53RTG* copy number and body size evolution

Relative divergence (duplication) times of the TP53 retrogenes were estimated using the TP53 alignment described above and BEAST (v1.7.4) (*Rohland et al., 2010*). We used the general time reversible model (GTR), empirical nucleotide frequencies (+F), a proportion of invariable sites estimated from the data (+I), four gamma distributed rate categories (+G), an uncorrelated lognormal relaxed molecular clock to model substitution rate variation across lineages, a Yule speciation tree prior, uniform priors for the GTR substitution parameters, gamma shape parameter, proportion of invariant sites parameter, and nucleotide frequency parameter. We used an Unweighted Pair Group Arithmetic Mean (UPGMA) starting tree.

To obtain posterior distributions of estimated divergence times, we use five node calibrations modeled as normal priors (standard deviation = 1) to constrain the age of the root nodes for the Eutheria (104.7 MYA), Laurasiatheria (87.2 MYA), Boreoeutherian (92.4 MYA), Atlantogenatan (103 MYA), and Paenungulata (64.2 MYA); divergence dates were obtained from www.timetree.org using the 'Expert Result' divergence dates. The analysis was run for 5 million generations and sampled every 1000 generations with a burn-in of 1 million generations; convergence was assessed using Tracer, which indicated convergence was reached rapidly (within 100,000 generations). Proboscidean body size data were obtained from previously published studies on mammalian body size evolution (*Evans et al., 2012*).

## Inference of *TP53/TP53RTG* copy number expansion through segmental duplications

We also observed that the genomic region surrounding each *TP53RTG* gene contained blocks of homolgous transposable element insertions, suggesting that these regions are segmental duplications. To confirm this observation, we used MUSCLE (*Edgar, 2004*) to align an approximately 20 kb region surrounding each *TP53RTG* gene and found that conservation within this region was very high, again suggesting these regions are relatively recent segmental duplications. To identify if the contigs on which *TP53RTG* genes are located contained locally collinear blocks (LCBs), as expected

for segmental duplications, we aligned contigs using progressiveMAUVE (*Darling et al., 2004*) as implemented in Genious (v6.1.2).

## Phylogenetic analyses of *TP53/TP53RTG* genes

We generated a dataset of *TP53* orthologs from 65 diverse mammals identified from GenBank, and included the *TP53* genes and retrogenes we identified from the African elephant, hyrax, manatee, tenrec, cape elephant shrew, and armadillo genomes. Nucleotide sequences were aligned using the MAFFT algorithm (*Katoh and Standley, 2014*) and the FFT refinement strategy implemented in the GUIDANCE webserver (*Penn et al., 2010*). Alignment confidence was assessed with 100 bootstrap replicates and ambiguously aligned sites (under the default GUIDANCE exclusion rule) removed prior to phylogenetic analyses; lineage specific insertions and deletions were also removed prior to phylogenetic analyses.

TP53 phylogenies were inferred using maximum likelihood implemented in PhyML (v3.1) (*Guindon et al., 2010*) using a general time reversible model (GTR), empirical nucleotide frequencies (+F), a proportion of invariable sites estimated from the data (+I), four gamma distributed rate categories (+G), and using the best of NNI and SPR branch moves during the topology search. Branch supports were assessed using the aBayes, aLRT, SH-like, and Chi2-based 'fast' methods as well as 1000 bootstrap replicates (*Guindon et al., 2010*). We also used MrBayes (v3.2.2 x64) (*Huelsenbeck and Ronquist, 2001*) for Bayesian tree inference using the GTR+I+G model; the analysis was run with two simultaneous runs of four chains for 4 million generations with the chains sampled every 1000 generations. We used Tracer (v1.5) to assess when the chains reached stationarity, which occurred around generation 100,000. At completion of the run the PRSF value was 1.00 indicating the analyses had reached convergence.

## Gene expression data (RNA-Seq)

To determine if *TP53RTG* genes were expressed, we generated RNA-Seq data from term placental villus and adipose tissue from African elephants (*Loxodonta africana*) and primary dermal fibroblasts from Asian elephants (*Elepas maximus*). Briefly, sequencing libraries were prepared using standard Illumina protocols with poly(A) selection, and sequenced as 100 bp single-end reads on a HiSeq2000. We also used previously published RNA-Seq data generated from primary fibroblasts (isolated from ear clips) from a male (GSM1227965) and female (GSM1227964) African elephant generated on an Illumina Genome Analyzer IIx (101 cycles, single end) and a male (GSM1278046) African elephant generated on an Illumina HiSeq 2000 (101 cycles, single end) (*Cortez et al., 2014*) and combined these reads into a single dataset of 138,954,285 reads. Finally, we used previously published 100 bp single-end reads generated on a HiSeq2000 to identify *TP53RTG* transcripts from Asian elephant PBMCs (SRX1423033) (*Reddy et al., 2015*).

Reads were aligned to a custom built elephant reference gene set generated by combining the sequences of the canonical *TP53* gene and *TP53RTG* genes with the ENSEMBL African elephant (*Loxodonta africana*) CDS gene build (Loxodonta_africana.loxAfr3.75.cds.all.fa) with Bowtie2 (*Langmead and Salzberg, 2012*). Bowtie two settings were: (1) both local alignment and end-to-end mapping; (2) preset option: sensitive; (3) Trim n-bases from 5' of each read: 0; and (4) Trim n-bases from 3' of each read: 0. Transcript assembly and FPKM estimates were generated with Cufflinks (version 0.0.7) (*Trapnell et al., 2012*) using aligned reads from Bowtie2, non-default parameters included quartile normalization and multi-read correction. Finally, we transformed FPKM estimates into transcripts per million (TPM), TPM=(FPKM per gene/sum FPKM all genes)x$10^6$, and defined genes with TPM $\geq$ 2 as expressed (*Li et al., 2010*; *Wagner et al., 2012*, *2013*).

The TP53RTG genes are 80.0–82.7% identical to TP53 at the nucleotide level, with 204–231 total nucleotide differences compared to TP53. This level of divergence allows for many reads to be uniquely mapped to each gene, there will also be significant read mapping uncertainty in regions of these genes with few nucleotide differences. However, if read mapping uncertainty was leading to false positive mappings of *TP53* derived reads to *TP53RTG* genes we would expect to observe the expression of many *TP53RTG* genes, rather the robust expression of a single (*TP53RTG12*) gene. We also counted the number of uniquely mapped reads to each *TP53RTG* gene and *TP53*. We found that 0–8 reads were uniquely mapped to most TP53RTG genes, except *TP53RTG12* which had ~115 uniquely mapped reads across samples and TP53 which had ~3000 uniquely mapped reads across

tissue samples. Thus we conclude that read mapping uncertainty has not adversely affected our RNA-Seq analyses.

## Gene expression data (PCR and Sanger sequencing)

We further confirmed expression of *TP53RTG12* transcripts in elephant cells through RT-PCR, taking advantage of differences between *TP53RTG12* and *TP53* sequences to design two aligned primer sets, one TP53-specific, the other *TP53RTG12* specific. *TP53RTG12* primers were: 5′ *ggg gaa act cct tcc tga ga* 3′ (forward) and 5′ *cca gac aga aac gat agg tg* 3′ (reverse). TP53 primers were: 5′ *atg gga act cct tcc tga ga* 3′ (forward) and 5′ *cca gac gga aac cat agg tg* 3′ (reverse). The *TP53* amplicon is expected to be 251 bps in length, while deletions present in the *TP53RTG12* sequence lead to a smaller projected amplicon size of 220 bps.

Total RNA was extracted from cultured *Loxodonta* and *Elephas* fibroblasts (RNAeasy Plus Mini kit, Qiagen), then DNase treated (Turbo DNA-free kit, Ambion) and reverse-transcribed using an olgio-dT primer for cDNA synthesis (Maxima H Minus First Strand cDNA Synthesis kit, Thermo Scientific). Control RT reactions were otherwise processed identically, except for the omission of reverse transcriptase from the reaction mixture. RT products were PCR-amplified for 45 cycles of 94°/20 s, 56°/30 s, 72°/30 s using a BioRad CFX96 Real Time qPCR detection system and SYBR Green master mix (QuantiTect, Qiagen). PCR products were electrophoresed on 3% agarose gels for 1 hr at 100 volts, stained with SYBR safe, and imaged in a digital gel box (ChemiDoc MP, BioRad) to visualize relative amplicon sizes. PCR products were also directly sequenced at the University of Chicago Genomics core facility, confirming projected product sizes and sequence identities.

## Gene prediction, promoter annotation, and TP53RTG transcript identification

We used geneid v1.2 (http://genome.crg.es/software/geneid/geneid.html) to infer if the *TP53RTG12* gene contained a non-coding exon 5′ to the predicted ATG start codon. For gene structure prediction we used the full-length scaffold_825 from African elephant (Broad/loxAfr3) sequence and forced an internal exon where the *TP53RTG12* gene is encoded in scaffold_825. Geneid identified a putative exon 5′ to the *TP53RTG12* gene from nucleotides 1761–1935 and an exon 3′ from nucleotides 6401–6776. We also used GENESCAN (http://genes.mit.edu/cgi-bin/genscanw_py.cgi) to predict the location of exons in the full-length scaffold_825 sequence and identified a putative exon from nucleotides 1750–1986. We next used Bowtie2 (*Langmead and Salzberg, 2012*) to map African and Asian elephant fibroblast RNA-Seq data onto African elephant scaffold_825 with the default settings.

## Taxonomic distribution of the RTE1_LA retrotransposon

The RTE1_LA non-LTR retrotransposon has previously been described from the African elephant genome, these elements are generally more than 90% identical to the consensus RTE1_LA sequence but less than 70% identical to other mammalian RTEs (http://www.girinst.org/2006/vol6/issue3/RTE1_LA.html). These data suggest that the RTE_LA element has relatively recently expanded in the elephant genome. To determine the taxonomic distribution of the RTE1_LA element we used BLAT to search the lesser hedgehog tenrec (*Echinops telfairi*), rock hyrax (*Procavia capensis*), West Indian manatee (*Trichechus manatus*), armadillo (*Dasypus novemcinctus*), and sloth (*Choloepus hoffmanni*) genomes. We identified numerous copies of the RTE_LA element in the genome of the Afrotherians, but not the Xenarthran genomes. These data indicate that the RTE_LA element is Afrotherian-specific, rather than Elephant-specific.

## TP53/TP53RTG western blotting

Elephant and Hyrax primary fibroblasts (San Diego Zoo, 'Frozen Zoo') were grown to confluency in 10 cm dishes at 37°C/5% $CO_2$ in a culture medium consisting of FGM/EMEM (1:1) supplemented with insulin, FGF, 6% FBS and Gentamicin/Amphotericin B (FGM-2, singlequots, Clonetics/Lonza). Culture medium was removed from dishes just prior to UV treatment and returned to cells shortly afterwards. Experimental cells were exposed to 50 J/m²UV-C radiation in a crosslinker (Stratalinker 2400, Stratagene), while control cells passed through media changes but were not exposed to UV. A small volume (~3 mL) of PBS covered fibroblasts at the time of UV exposure. To inhibit TP53

proteolysis, MG-132 (10 µM) was added to experimental cell medium 1 hr prior to UV exposure, and maintained until the time of cell lysis. 5 hr post-UV treatment, cells were briefly rinsed in PBS, then lysed and boiled in 2x SDS-PAGE sample buffer. Lysates were separated via SDS-PAGE on 10% gels for 1 hr at 140 volts, then electrophoretically transferred to PVDF membranes (1 hr at 85 volts). Membranes were blocked for 1 hr in 5% milk in TBST and incubated overnight at 4°C with rabbit polyclonal TP53 antibodies (FL-393, Santa Cruz Biotechnology, and ab131442, Abcam). Blots were washed 3x in TBST, incubated with HRP-conjugated, anti-rabbit IgG 2° antibodies for 1 hr at RT, and washed four more times in TBST. Protein bands were visualized via enhanced chemiluminescence (BioRad Clarity), and imaged in a digital gel box (Chemidoc MP, BioRad). Western blots were replicated three independent times.

## Co-immunoprecipitation

Human HEK-293 cells were grown to 80% confluency in 20 cm dishes at 37°C/5% $CO_2$ in a culture medium consisting of DMEM supplemented with 10% FBS. At 80% confluency, cells were transiently transfected with the TP53RTG12 pcDNA3.1(+)/myc-His expression vector. After 16 hr the transfection media was removed and replaced with fresh DMEM, and the cells were incubated an additional 24 hr before harvesting. After removing DMEM and washing cells twice with PBS, 1 mL ice-cold lysis buffer (20 mM Tris, pH 8.0, 40 mM KCl, 10 mM MgCl2, 10% glycerol, 1% Triton X-100, 1x Complete EDTA-free protease inhibitor cocktail (Roche), 1x PhosSTOP (Roche)) was added to each plate and cells were harvested by scraping with a rubber spatula. Cells were then incubated on ice for 30 min in 420 mM NaCl. The whole cell lysate was cleared by centrifugation at 10,000 rpm for 30 min at 4°C, and the supernatant was transferred to a clean microfuge tube. After equilibrating protein concentrations, 1 mL of sample was mixed with 40 mL of α-MDM2 or α-Myc antibody conjugated agarose beads (Sigma) pre-washed with TNT buffer (50 mM Tris-HCl, pH 7.5, 150 mM NaCl, 0.05% Triton X-100), and rotated overnight at 4°C. The following day, samples were treated with 50 U DNase (Roche) and 2.5 µg RNase (Roche) for 60 min at room temperature, as indicated. Samples were washed 3x with 1 mL wash buffer (150 mM NaCl, 0.5% Triton X-100). After the final wash, agarose beads were resuspended in elution buffer (500 mM Tris pH 7.5, 1 M NaCl), and boiled to elute immunoprecipitated complexes. Eluted protein was run on Bis-tris gels, probed with antibodies and visualized by Chemi-luminescence. Serial Westerns were performed for each antibody following chemical stripping and re-blocking. Antibodies were from Santa Cruz: MDM2 (SMP14) sc-965, lot #J2314; p53 (Fl-393) sc-6243, lot # D0215; c-Myc (9E10) sc-40, lot # G2413.

## ApoTox-Glo viability/cytotoxicity/apoptosis analyses

Primary fibroblasts were grown to 80% confluency in T-75 culture flasks at 37°C/5% $CO_2$ in a culture medium consisting of FGM/EMEM (1:1) supplemented with insulin, FGF, 6% FBS and Gentamicin/Amphotericin B (FGM-2, singlequots, Clonetics/Lonza). $10^4$ cells were seeded into each well of an opaque bottomed 96-well plate, leaving a column with no cells (background control); each 96-well plate contained paired elephant and hyrax samples. Serial dilutions of Doxorubicin (0 uM, 0.5 uM, 1.0 uM, 5 uM, 10 uM and 50 uM), Mitomycin c (0 uM, 0.5 uM, 1.0 uM, 5 uM, 10 uM and 50 uM), and Nutlin-3a (0 uM, 0.5 uM, 1.0 uM, 5 uM, 10 uM and 50 uM) and 90% culture media were added to each well such that there were four biological replicates for each condition. After 18 hr of incubation with each drug, cell viability, cytotoxicity, and caspase-3/7 activity were measured using the Apo-Tox-Glo Triplex Assay (Promega) in a GloMax-Multi+ Reader (Promega). Data were standardized to no-drug control cells. For UV-C treatment, culture medium was removed from wells prior to UV treatment and returned to cells shortly afterwards. Experimental cells were exposed to 50 J/m$^2$UV-C radiation in a crosslinker (Stratalinker 2400, Stratagene), while control cells passed through media changes but were not exposed to UV. A small volume (~30 uL) of PBS covered fibroblasts at the time of UV exposure. Cell viability, cytotoxicity, and caspase-3/7 activity were measured using the ApoTox-Glo Triplex Assay (Promega) in a GloMax-Multi+ Reader (Promega) 6, 12, 28.5 and 54.5 hr after UV-C treatment. Data were standardized to no UV-C exposure control cells. ApoTox-Glo Triplex Assays were replicated three independent times.

## Luciferase assays

Primary fibroblasts were grown to 80% confluency in T-75 culture flasks at 37°C/5% $CO_2$ in a culture medium consisting of FGM/EMEM (1:1) supplemented with insulin, FGF, 6% FBS and Gentamicin/ Amphotericin B (FGM-2, singlequots, Clonetics/Lonza). At confluency, cells were trypsinized, centrifuged at 90 g for 10 min and resuspended in nucleofection/supplement solution and incubated for 15 min. After incubation 1ug of the pGL4.38[*luc2p*/p53 RE/Hygro] luciferase reporter vector and 100 ng of the pGL4.74[*hRluc*/TK] Renilla reporter vector were transiently transfected into an elephant and hyrax cells using the Amaxa Basic Nucleofector Kit (Lonza) using protocol T-016. Immediately following nuleofection, $10^4$ cells were seeded into each well of an opaque bottomed 96-well plate, leaving a column with no cells (background control); each 96-well plate contained paired elephant and hyrax samples. 24 hr after nucleofection cells were treated with either vehicle control, Doxorubicin, Mitomycin c, Nutlin-3a, or 50 J/m$^2$UV-C. Luciferase expression was assayed 18 hr after drug/ UV-C treatment cells, using the Dual-Luciferase Reporter Assay System (Promega) in a GloMax-Multi+ Reader (Promega). For all experiments luciferase expression was standardized to Renilla expression to control for differences nucleofection efficiency across samples; Luc./Renilla data is standardized to (Luc./Renilla) expression in untreated control cells. Each luciferase experiment was replicated three independent times.

## siRNA experiments

siRNAs designed to specifically-target TP53RTG were validated via qRT-PCR using the two primer sets described earlier, which amplify either TP53RTG or canonical TP53 cDNAs, to confirm specificity and efficacy of knockdown. Sequences of the three TP53RTG-specific siRNAs used are as follows: (1) 5'-CAGCGGAGGCAGUAGAUGAUU-3', (2) 5'-GGCUCAAGGAAUAUCAGAAUU-3', (3) 5'-CAG-CAGCGGAGGCAGUAGAUU-3' (Dharmacon). Loxodonta fibroblasts were transfected with siRNAs using Lipofectamine LTX, and tested 48–72 hr later for either TP53 response via luciferase assay or for cell viability/toxicity/apoptosis via ApoTox-Glo assay.

## Acknowledgements

We acknowledge the financial support of The University of Chicago (VJL) and the University of Nottingham (NPM, LY, RDE). RDE was additionally supported by funding through the Advanced Data Analysis Centre. African Elephant Adipose tissue samples used for mRNA-seq were kindly provided through collaboration with T Allen (The Paul Mellon Laboratory, Newmarket, Suffolk, United Kingdom) and F Stansfield (The Elephant Research and Conservation Unit, Savé Valley Conservancy, Harare, Zimbabwe).

## Additional information

### Funding

| Funder | Author |
| --- | --- |
| University of Nottingham | Lisa Yon<br>Nigel P Mongan<br>Richard D Emes |
| Advanced Data Analysis Centre | Richard D Emes |
| University of Chicago | Vincent J Lynch |

The funders had no role in study design, data collection and interpretation, or the decision to submit the work for publication.

### Author contributions

MS, Conception and design, Acquisition of data, Analysis and interpretation of data, Drafting or revising the article, Performed experiments; LF, KM, Conception and design, Acquisition of data, Analysis and interpretation of data, Performed experiments; SC, Conception and design, Acquisition of data, Performed experiments; LY, NPM, RDE, Analysis and interpretation of data, Drafting or

revising the article, Contributed unpublished essential data or reagents, Performed experiments; VJL, Conception and design, Analysis and interpretation of data, Drafting or revising the article

**Author ORCIDs**

Vincent J Lynch, http://orcid.org/0000-0001-5311-3824

## Additional files

### Major datasets

The following dataset was generated:

| Author(s) | Year | Dataset title | Dataset URL | Database, license, and accessibility information |
|---|---|---|---|---|
| Sulak M, Fong L, Mika K, Chigurupati S, Yon L, Mongan NP, Emes RD, Lynch VJ | 2016 | Data from: TP53 copy number expansion is associated with the evolution of increased body size and an enhanced DNA damage response in elephants | http://dx.doi.org/10.5061/dryad.968vr | Available at Dryad Digital Repository under a CC0 Public Domain Dedication |

The following previously published datasets were used:

| Author(s) | Year | Dataset title | Dataset URL | Database, license, and accessibility information |
|---|---|---|---|---|
| Dastjerdi A, Robert C, Watson M | 2014 | Whole genome sequencing of two Asian elephants (Elephas maximus) infected with Elephant Endotheliotropic Herpesvirus (EEHV) | http://www.ncbi.nlm.nih.gov/bioproject/PRJEB4905 | Publicly available at the NCBI Short Read Archive (accession no: ERX334765, ERX334764). |
| Enk J, Rouillard J-M, Poinar H | 2013 | Raw Illumina read files associated with Enk et al. publication on qPCR as a predictor of mapped reads after enrichment. Sequence data from non-enriched and enriched libraries deriving from Pleistocene-age bone and tooth remains of various Mammuthus sp. | http://www.ncbi.nlm.nih.gov/bioproject/PRJNA203139 | Publicly available at the NCBI Short Read Archive (accession no: SRX329134, SRX329135, SRX327587, SRX327586, SRX327583, SRX327582). |
| Rohland N, Reich D, Mallick S, Meyer M, Green RE, Georgiadis NJ, Roca AL, Hofreiter M | 2010 | Mastodon shotgun sequencing | http://www.ncbi.nlm.nih.gov/sra?term=SRP001730 | Publicly available at the NCBI Short Read Archive (accession no: SRX015822, SRX015823). |
| Cortez D, Marin R, Toledo-Flores D, Froidevaux L, Liechti A, Waters PD, Grützner F, Kaessmann H | 2013 | Origins and functional evolution of Y chromosomes across mammals | http://www.ncbi.nlm.nih.gov/bioproject/PRJNA218629 | Publicly available at the NCBI Short Read Archive (accession no: GSM1227965, GSM1227964). |
| Reddy PC, Sinha I, Kelkar A, Habib F, Pradhan SJ, Sukumar R, Galande S | 2015 | Comparative sequence analyses of genome and transcriptome reveal novel transcripts and variants in the Asian elephant Elephas maximus | http://www.ncbi.nlm.nih.gov/bioproject/PRJNA301482 | Publicly available at the NCBI Short Read Archive (accession no: SRX1423033). |

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
