## [Decision Letter]

Thank you for submitting your work entitled "TP53 duplication correlates with the evolution of increased body size and an enhanced DNA damage response in elephants" for consideration by *eLife*. Your article has been favorably evaluated by Diethard Tautz as the Senior Editor and three reviewers, one of whom is a member of our Board of Reviewing Editors.

The reviewers have discussed the reviews with one another and the Reviewing editor has drafted this decision to help you prepare a revised submission.

The reviewers discussed the manuscript at length, and found it to be very interesting and likely to become a major contribution to the field of evolutionary medicine. One of the reviewers pointed out that some of the findings in the manuscript were likely to become material for biology textbooks. Another Reviewer described the findings as 'fascinating' and 'inspiring'.

The manuscript describes the interesting fact that the tumor suppressor gene TP53 has undergone significant copy number gain in the elephant (Proboscidean) lineage, which may solve the so-called Peto's paradox, i.e. the lack of correlation between body size/lifespan and tumor incidence in the animal kingdom. The manuscript documents several interesting findings, including:

a) Expansion of the p53 gene via segmental duplication of 'retrogenes'.

b) Existence of a promoter sequence enabling transcription of these additional copies.c) Expression analysis of the retrogenes via RNAseq and PCR.

d) Increased protein expression of 'retroproteins' after DNA damage by UVC.

e) Increased p53 reporter activity and apoptosis in response to diverse p53 activating agents.

f) Evidence of a 'transdominant' effect, whereas the truncated protein 'retroproteins' may enhance p53 activity by protecting it from MDM2-mediated repression.

The reviewers discussed the fact that during the review process a similar paper was published by the Schiffman group. However, the reviewers concluded that the current manuscript by Sulak et al. contains significantly more information, and that it also reaches somewhat different conclusions about the expression and mechanism of action of the p53 retrogenes. Accordingly, the reviewers agreed to recommend submission of a revised manuscript that would not only address several concerns about experimental design and data analysis, but that would also contain an exhaustive comparison and discussion of the findings in the two papers.

The key concerns raised by reviewers are:

A) Comparison and discussion of the findings in the current manuscript versus those in the paper by the Schiffman team. Reviewers concluded that it would be critical for readers to clearly understand the additional information in this paper, as well as the areas of conflicting observations that may require future investigation. The paper from Schiffman et al. needs to be cited and acknowledged in the current manuscript. Reviewers observed that since this paper complements and extends the published work in significant ways, it may well end up being more highly cited, and that proper citation practice would be to cite both of these together. Some examples of the aspects that should be compared between the two papers are:

A1) The current manuscript contains an elegant demonstration of the mechanism of copy number gain via segmental duplication, which should be highlighted. The results described in the first three subsections of the Results section, on the phylogenetic origin and the association of increased copy number with increased body size provide independent added value that goes beyond the Schiffman.

A2) Some of these results on RTG expression are more precise than those reported in the Schiffman et al. paper. The authors need to go over the supplementary material as well as the results in the published paper and point out where they agree, where they do not, and where these results extend the results from Schiffman et al. For example, using RNA-seq, the authors show that the retrogene RTG12 is the variant most predominantly transcribed in elephant cells. In contrast, Schiffman et al. employed RT-PCR to document expression of 'retrogenes'. How many of the retrogenes are actually expressed? Is it possible that retrogenes are expressed in a cell type-specific manner (e.g. fibroblasts in this paper versus lymphoblastoids in Schiffman's paper?).

B) Unclear mechanism of action of the retrogene-encoded p53 variants. The two papers disagree on the mechanism of action of the retrogene-encoded proteins. Here, the authors describe a 'transdominant' effect that does not involve MDM2 binding. In contrast, the work by Schiffman et al. concludes that the retroproteins do bind to MDM2. The data from both papers is weak in this regard. If RTG12 does not bind to MDM2 as indicated in this paper, how is then its expression controlled upon DNA damage? What is the mechanism by which UVC leads to significantly more expression of RTG12? To address this issue, the authors should test whether RTG12 induction is at the level of RNA or protein stability, and to define whether Nutlin (an inhibitor of the MDM2-p53 interaction) also induces protein accumulation in this setting. Furthermore, their current model does not include stabilization of RTG12 protein upon DNA damage. How does RTG12 in this manuscript compare to RTG9 in the paper by Schiffman et al? Could it be that differences in putative binding to MDM2 among the two papers are due to sequence variations in the N-terminus domain of the various retroproteins? To address this, the reviewers request a figure showing aminoacid and nucleotide alignments of the N-terminus transactivation domain regions for all RTGs, highlighting the residues known to be important for transactivation and binding to MDM2, and including all proper Genbank identifiers for each sequence. One of the reviewers pointed out that the MDM2 binding site is mutated in all the TP53RTG proteins (for those that initiate at the first ATG). Furthermore, the same Reviewer observed that, from analysis of the sequences available in Genbank, all the TP53RTGs except one can produce a protein of at least 79 amino-acids, as some TP53RTG can be translated from codon 728 or 940. This should be discussed in the revised manuscript. To further support their 'transdominant' mechanism, the authors should simply perform co-immunoprecipitation of full-length p53 protein using the C-terminal p53-specific 421 monoclonal antibody (the epitope is conserved in the elephant p53 protein). The 421 antibody will not bind to TP53RTG proteins, which lack the 421 epitope. Thus, the co-immunoprecipitation of TP53RTG proteins with full-length elephant p53 protein will unequivocally prove the formation of a protein complex with full-length p53. Finally, the authors propose that the TP53RTG proteins inhibit tetramerization of p53. However, tetramerization of p53 is absolutely required for p53 transcriptional activity. In addition, MG132 seems to induce ubiquitination of p53 in elephant cells treated with DNA-damaging agents despite expression of TP53RTG12. Therefore, the model involving inhibition of tetramerization and ubiquitination is not possible. Altogether, the experiments proposed here will enable the authors to produce a more accurate model of p53 regulation by the p53 retroproteins in elephant cells.

C) Concern about the use overexpression experiments and HEK293 cells. Reviewers were concerned about the use of ectopic overexpression of TP53RTG12 in mouse cells and human HEK293 cells. The observed phenotypes (e.g. hypersensitivity to DNA damage) could be due to artifact-prone overexpression analysis. In addition, it is well established that p53 is inactivated and stabilized by a viral transforming protein in HEK293 cells. Therefore, the value of the MDM2 experiments conducted in HEK293 cells is unclear. In order to unequivocally demonstrate that TP53RTGs regulate the p53-dependent cellular response to DNA damage, the authors must knockdown endogenous TP53RTG expression using TP53RTG-specific siRNAs in elephant cells. Since the authors have identified long nucleotide sequences unique to TP53RTG mRNAs, it is then possible to design siRNAs that will reduce TP53RTG expression without affecting TP53 expression in elephant cells. It will be possible then for the authors to test their hypothesis in elephant cells. Furthermore, transfection of TP53RTG siRNA would enable the authors to unequivocally identify the protein bands corresponding to TP53RTG proteins via Western blot.

D) Interpretation of the various 'protein bands' observed in elephant cells. In addition to the siRNA experiments mentioned above, reviewers pointed out that the authors cannot accurately identify and name the p53 isoforms observed in their electrophoresis experiments. The elephant p53 protein has no methionine at codon133 and p53psy has not been demonstrated in elephant cells. Furthermore, the bands above 50kD may be polymers or ubiquitinated p53 proteins or cross-reacting bands. Thus, it is essential to define the identity of these bands with TP53 siRNAs (the C-terminal region would provide ample sequence space to design these siRNAs). Also, the protein molecular weight marker is not displayed in any of the immunoblots, which prevents proper interpretation of these results.

E) Incomplete discussion of the impact of p53 copy number gain in organismal biology, tumor suppression and ageing/senescence. The p53 literature abounds with studies exploring the impact of additional copies of the p53 gene, including 'retrogenes'. The reviewers concluded that this manuscript would be much improved by a thorough discussion of these studies. For example, the authors should discuss the p53 pseudogenes in mouse and rat, which are expressed and seem to compromise the activity of functional p53. Importantly, it is well established that an extra copy of full-length p53 in mice leads to hyperactive apoptotic activity, shortening the lifespan of the organism and also inducing neurodegenerative diseases. Thus, the authors should discuss the 'super p53' mouse models (García-Cao et al., EMBO J 2002; Tyner et al., Nature 2002; Maier et al. Genes and Dev 2004).

Another important point in this area is that the authors have not discussed the fact that TP53RTG12 seems to be the only highly expressed p53 pseudogene in elephant cells. How could the authors conclude that the low cancer incidence in elephants is due to the 'multiple copies' of p53 pseudogenes, while the effects of all TP53RTGs seem to be driven by a single retrogene, TP53RTG12? In fact, the authors show that all the TP53RTGs are expressed in the RNA seq data -although at very variable levels-, and they should emphasize this observation; otherwise the story does not reconcile with the existence of two expressed p53 pseudogenes in rat and the two or single expressed p53 pseudogenes in mouse (Tanooka et al., Cancer Res. 1998 and Gene 2001). The authors made a very important discovery, as elephant species may have evolved a molecular mechanism to uncouple the tumor suppressive activity of p53 from its pro-ageing activity. Therefore, the authors should elaborate on whether it is the number of RTGs and/or the tissue specific expression of RTGs and/or the type of RTG (i.e. as defined by the amino-acid sequence of the RTG) that fine tunes p53 activities toward enhanced tumor suppression without hyper-reactivity to cellular stress.

---

## [Author Response]

*The reviewers discussed the fact that during the review process a similar paper was published by the Schiffman group. However, the reviewers concluded that the current manuscript by Sulak et al. contains significantly more information, and that it also reaches somewhat different conclusions about the expression and mechanism of action of the p53 retrogenes. Accordingly, the reviewers agreed to recommend submission of a revised manuscript that would not only address several concerns about experimental design and data analysis, but that would also contain an exhaustive comparison and discussion of the findings in the two papers.*

We thank the editors for the opportunity to revise and resubmit our manuscript.

We have modified the main text, specifically the Results and Discussion sections, to address reviewer comments including the addition of several new experiments, and expanded details of results to clarify reviewer concerns and misunderstandings. Among the new experiments are a demonstration that TP53RTG genes are necessary for the augmented TP53 signaling response in elephants via siRNA-mediated knock-down of TP53RTG transcripts and a detailed structural and functional comparison of our model of TP53RTG mechanism of action vs the model proposed by the Abegglen et al. (2015). We elaborate on these additional experiments as well as other changes below.

*The key concerns raised by reviewers are:*

*A) Comparison and discussion of the findings in the current manuscript versus those in the paper by the Schiffman team. Reviewers concluded that it would be critical for readers to clearly understand the additional information in this paper, as well as the areas of conflicting observations that may require future investigation. The paper from Schiffman et al. needs to be cited and acknowledged in the current manuscript. Reviewers observed that since this paper complements and extends the published work in significant ways, it may well end up being more highly cited, and that proper citation practice would be to cite both of these together. Some examples of the aspects that should be compared between the two papers are:*

*A1) The current manuscript contains an elegant demonstration of the mechanism of copy number gain via segmental duplication, which should be highlighted. The results described in the first three subsections of the Results section, on the phylogenetic origin and the association of increased copy number with increased body size provide independent added value that goes beyond the Schiffman.*

*A2) Some of these results on RTG expression are more precise than those reported in the Schiffman et al. paper. The authors need to go over the supplementary material as well as the results in the published paper and point out where they agree, where they do not, and where these results extend the results from Schiffman et al. For example, using RNA-seq, the authors show that the retrogene RTG12 is the variant most predominantly transcribed in elephant cells. In contrast, Schiffman et al. employed RT-PCR to document expression of 'retrogenes'. How many of the retrogenes are actually expressed? Is it possible that retrogenes are expressed in a cell type-specific manner (e.g. fibroblasts in this paper versus lymphoblastoids in Schiffman's paper?).*

We apologize for not explicitly addressing the Abegglen et al. (2015) study in our prior submission, although we were aware of their manuscript was forthcoming we did not have access to it prior to our submission to *eLife* (it was published a few days after our submission).

In our revised submission we have dedicated a section of the Discussion (‘Comparison to previous studies of elephant TP53’) to specifically compare and contrast our findings to those of Abegglen et al. (2015). While we compare our results to Abegglen et al. (2015) in detail in the revised Discussion, our key observations are that: 1) Abegglen et al. (2015) did not determine the mechanisms of expansion whereas we have, therefore we have elaborated our discussion of the mechanisms of duplication; 2) Abegglen et al. (2015) found that elephant and human cells had different sensitivities to ionizing radiation, but their taxon sampling did not allow for polarizing which species was different whereas our explicit phylogenetic approach allows us to determine that elephants evolved increased sensitivity to DNA-damage; and 3) Abegglen et al. (2015) used RT-PCR and Sanger sequencing to show two distinct transcripts were expressed in elephant PBMCs, but they did not assign the loci to which these transcripts correspond. We analyzed the chromatograms shown in Abegglen et al. eFigure 4 and found that the 185bp product is a transcript from the TP553RTG14 gene and the 201bp product is a transcript from the TP553RTG5 gene. Thus, our combined data suggest that at least five TP53RTG genes are transcribed. Furthermore, we did not observe TP553RTG14 or TP553RTG5 expression in adipose, placenta, or fibroblasts suggesting that the expression of some TP53RTG genes is tissue-specific.

*B) Unclear mechanism of action of the retrogene-encoded p53 variants. The two papers disagree on the mechanism of action of the retrogene-encoded proteins. Here, the authors describe a 'transdominant' effect that does not involve MDM2 binding. In contrast, the work by Schiffman et al. concludes that the retroproteins do bind to MDM2. The data from both papers is weak in this regard. If RTG12 does not bind to MDM2 as indicated in this paper, how is then its expression controlled upon DNA damage? What is the mechanism by which UVC leads to significantly more expression of RTG12? To address this issue, the authors should test whether RTG12 induction is at the level of RNA or protein stability, and to define whether Nutlin (an inhibitor of the MDM2-p53 interaction) also induces protein accumulation in this setting. Furthermore, their current model does not include stabilization of RTG12 protein upon DNA damage.*

We have greatly expanded our revised manuscript and now dedicate a section of Results, a figure (Figure 10), and a section of the Discussion to comparing our model to the Abegglen et al. (2015) model and specifically test our model and theirs. While those analyses are too complicated to include here, we show that: 1) Abegglen et al. (2015) actually do not demonstrate an interaction between TP53RTG9 and MDM2, the co-IP shown in their eFigure 5C *does not* have a band at the size expected for MDM2 (eFigure 5C, top panel). It is unclear to us why they conclude there MDM2 co-IP’d with TP53RTG9; 2) A tryptophan residue in TP53 critical for the interaction between TP53 and MDM2 is substituted for glycine in all TP53RTG proteins; 3) This W->G mutation is predicted to severely destabilize the TP53RTG/MDM2 interaction; and 4) TP53RTG12 does not co-IP with MDM2. Thus we conclude that the Abegglen et al. (2015) model of TP53RTG proteins acting as decoys for the MDM complex is unlikely.

In contrast we show that: 1) The TP53 dimerization domain in TP53RTG proteins is conserved; 2) A model of the TP53/TP53RTG12 dimer is nearly indistinguishable from the TP53/TP53 dimer; and 3) TP53RTG12 co-IPs with TP53. Based on these data we propose a model in which TP53RTG proteins dimerize with TP53, thereby protecting TP53 from the MDM complex because the MDM complex only (efficiently) binds to TP53 tetramers. We have yet to demonstrate how TP53 is released from the protection of TP53RTG proteins, clearly this is an important part of our model but given the scope of the current paper believe this is best left to a follow up paper.

We note, however, that we do not provide evidence that TP53RTG expression is increased upon UVC exposure. Our Western blot demonstrating bands at the expected size for TP53RTG12 (and TP53RTG19) is of total protein from cells treated with UVC and the proteasome inhibitor MG132. The goal of this experiment was to ascertain if there was evidence for TP53RTG proteins, not their regulation by DNA-damage. Thus we sought to stabilize TP53 and prevent its degradation, so that we might accumulate and potentially observe low abundance proteins. We agree that it is important to demonstrate how TP53RTG expression levels are affected by DNA damage, and if this is mechanism that releases TP53 from TP53RTG dimerization.

*How does RTG12 in this manuscript compare to RTG9 in the paper by Schiffman et al? Could it be that differences in putative binding to MDM2 among the two papers are due to sequence variations in the N-terminus domain of the various retroproteins? To address this, the reviewers request a figure showing aminoacid and nucleotide alignments of the N-terminus transactivation domain regions for all RTGs, highlighting the residues known to be important for transactivation and binding to MDM2, and including all proper Genbank identifiers for each sequence. One of the reviewers pointed out that the MDM2 binding site is mutated in all the TP53RTG proteins (for those that initiate at the first ATG). Furthermore, the same Reviewer observed that, from analysis of the sequences available in Genbank, all the TP53RTGs except one can produce a protein of at least 79 amino-acids, as some TP53RTG can be translated from codon 728 or 940. This should be discussed in the revised manuscript.*

We have compared the sequence of RTG9 from Abegglen et al. (2015) to our TP53RTG12 and conclude they are the same protein (100% protein and nucleotide identity). Thus based on our revised analyses of MDM2 binding by TP53RTG proteins, including the observation that the MDM2 biding site is mutated in all TP53RTG proteins (we thank the reviewer for this very helpful observation), we conclude that the Abegglen et al. (2015) model is not supported by current data. We now include a panel in Figure 9 showing the location of functional domains in TP53 and TP53RTG proteins, and logos of the MDM2 interaction motif and TP53 dimerization motif in TP53 and TP53RTG proteins to highly conserved and divergent residues, and a table that includes Ensembl IDs for all TP53RTG genes. We also briefly discussed alternative start sites for TP53RTG proteins in our conclusion section.

*To further support their 'transdominant' mechanism, the authors should simply perform co-immunoprecipitation of full-length p53 protein using the C-terminal p53-specific 421 monoclonal antibody (the epitope is conserved in the elephant p53 protein). The 421 antibody will not bind to TP53RTG proteins, which lack the 421 epitope. Thus, the co-immunoprecipitation of TP53RTG proteins with full-length elephant p53 protein will unequivocally prove the formation of a protein complex with full-length p53.*

This was a great suggestion and we worked for very many months to optimize our co-IP and Westerns with the 421 antibody, however, we have been unable to demonstrate that the 421 antibody recognizes the elephant TP53 protein. While this experiment would have been very informative, we have nonetheless demonstrated the formation of a protein complex with full-length human TP53 and Myc-tagged TP53RTG (Figure 10) in support of our transdominant model.

*Finally, the authors propose that the TP53RTG proteins inhibit tetramerization of p53. However, tetramerization of p53 is absolutely required for p53 transcriptional activity. In addition, MG132 seems to induce ubiquitination of p53 in elephant cells treated with DNA-damaging agents despite expression of TP53RTG12. Therefore, the model involving inhibition of tetramerization and ubiquitination is not possible. Altogether, the experiments proposed here will enable the authors to produce a more accurate model of p53 regulation by the p53 retroproteins in elephant cells.*

MG132 does not induce ubiquitination of TP53 in elephant cells treated with DNA-damaging agents despite expression of TP53RTG12, rather MG132 prevents the proteasome from degrading ubiquitinated proteins leading to their accumulation. Therefore, this experiment cannot be used to determine if TP53RTG proteins protect TP53 from ubiquitination. We have edited the text to clarify this point.

As the editors and reviewers can imagine, we are very eager to test our ‘guardian’ hypothesis and are currently developing an in vitro ubiquitination assay that will allow us to test if TP53RTG proteins protect TP53 from MDM2-mediated ubiquitination. While this hypothesis is a key part of our model, a full test is beyond the scope of the current manuscript. We now provide data indicating DNA damaging agents cause a rapid and extreme decrease in TP53RTG transcripts, suggesting down-regulation of TP53RTGs upon DNA damage allows for tetramerization of TP53 and the initiation of TP53 signaling.

*C) Concern about the use overexpression experiments and HEK293 cells. Reviewers were concerned about the use of ectopic overexpression of TP53RTG12 in mouse cells and human HEK293 cells. The observed phenotypes (e.g. hypersensitivity to DNA damage) could be due to artifact-prone overexpression analysis. In addition, it is well established that p53 is inactivated and stabilized by a viral transforming protein in HEK293 cells. Therefore, the value of the MDM2 experiments conducted in HEK293 cells is unclear. In order to unequivocally demonstrate that TP53RTGs regulate the p53-dependent cellular response to DNA damage, the authors must knockdown endogenous TP53RTG expression using TP53RTG-specific siRNAs in elephant cells. Since the authors have identified long nucleotide sequences unique to TP53RTG mRNAs, it is then possible to design siRNAs that will reduce TP53RTG expression without affecting TP53 expression in elephant cells. It will be possible then for the authors to test their hypothesis in elephant cells. Furthermore, transfection of TP53RTG siRNA would enable the authors to unequivocally identify the protein bands corresponding to TP53RTG proteins via Western blot.*

We agree that the most convincing demonstration of the functional importance of TP53RTG genes would be to knockdown or knockout their TP53RTG expression. We tried to knockdown expression with numerous TP53RTG-specific siRNAs targeting different regions of the TP53RTG transcripts before our initial submission. While these experiments were able to knockdown TP53RTG expression they always cross-reacted with TP53 leading to reduced TP53 expression as well.

Fortunately, we have recently identified a pan TP53RTG siRNA that reduces TP53RTG expression ~70% and does not cross react with TP53. To test if TP53RTGs are functional we used this siRNA to knockdown TP53RTG transcripts and assayed induction of the TP53 signaling pathway in response to DNA damage and MDM2 antagonism. We show that knockdown of TP53RTGs in untreated cells increased baseline TP53 signaling, consistent with our ‘Guardian’ model, and also reduced TP53 signal after treatment with DNA damage inducing agents and Nutlin-3a. These data suggest that TP53RTG proteins have two distinct functions, inhibiting TP53 signaling in the absence of inductive signals and potentiation of TP53 signaling after induction of DNA damage.

*D) Interpretation of the various 'protein bands' observed in elephant cells. In addition to the siRNA experiments mentioned above, reviewers pointed out that the authors cannot accurately identify and name the p53 isoforms observed in their electrophoresis experiments. The elephant p53 protein has no methionine at codon133 and p53psy has not been demonstrated in elephant cells. Furthermore, the bands above 50kD may be polymers or ubiquitinated p53 proteins or cross-reacting bands. Thus, it is essential to define the identity of these bands with TP53 siRNAs (the C-terminal region would provide ample sequence space to design these siRNAs). Also, the protein molecular weight marker is not displayed in any of the immunoblots, which prevents proper interpretation of these results.*

We agree that the data supporting translation of one or more TP53RTG genes is circumstantial and that the Western blot shown in Figure 5 is far from conclusive. We note that we do not conclude from this blot that TP53RTGs are translated, rather we conclude that we “identified an elephant-specific band at the expected size for the TP53RTG12 (19.6kDa) and TP53RTG19 (22.3kDa) proteins, suggesting that the *TP53RTG12* and *TP53RTG19* transcripts are translated in elephant fibroblasts” and explicitly state we identified “high molecular weight bands corresponding to previously reported SDS denaturation resistant TP53 oligomers (Cohen et al., 2008; Ottaggio et al., 2000) and (poly)ubiquitinated TP53 conjugates (Sparks et al., 2013)”.

Unfortunately identifying an antibody, particularly a monoclonal, that recognizes elephant TP53 and TP53RTGs has been remarkably difficult. However, our observation that knockdown of TP53RTGs and over-expression of TP53RTG12 in mouse cells has functional consequences on TP53 signaling and the induction of apoptosis provides further support that TP53RTG is translated and functional.

*E) Incomplete discussion of the impact of p53 copy number gain in organismal biology, tumor suppression and ageing/senescence. The p53 literature abounds with studies exploring the impact of additional copies of the p53 gene, including 'retrogenes'. The reviewers concluded that this manuscript would be much improved by a thorough discussion of these studies. For example, the authors should discuss the p53 pseudogenes in mouse and rat, which are expressed and seem to compromise the activity of functional p53. Importantly, it is well established that an extra copy of full-length p53 in mice leads to hyperactive apoptotic activity, shortening the lifespan of the organism and also inducing neurodegenerative diseases. Thus, the authors should discuss the 'super p53' mouse models (García-Cao et al., EMBO J 2002; Tyner et al., Nature 2002; Maier et al. Genes and Dev 2004).*

*Another important point in this area is that the authors have not discussed the fact that TP53RTG12 seems to be the only highly expressed p53 pseudogene in elephant cells. How could the authors conclude that the low cancer incidence in elephants is due to the 'multiple copies' of p53 pseudogenes, while the effects of all TP53RTGs seem to be driven by a single retrogene, TP53RTG12? In fact, the authors show that all the TP53RTGs are expressed in the RNA seq data -although at very variable levels-, and they should emphasize this observation; otherwise the story does not reconcile with the existence of two expressed p53 pseudogenes in rat and the two or single expressed p53 pseudogenes in mouse (Tanooka et al., Cancer Res. 1998 and Gene 2001). The authors made a very important discovery, as elephant species may have evolved a molecular mechanism to uncouple the tumor suppressive activity of p53 from its pro-ageing activity. Therefore, the authors should elaborate on whether it is the number of RTGs and/or the tissue specific expression of RTGs and/or the type of RTG (i.e. as defined by the amino-acid sequence of the RTG) that fine tunes p53 activities toward enhanced tumor suppression without hyper-reactivity to cellular stress.*

We have completely rewritten the Discussion to explicitly address the potential functional consequences of the TP53RTG genes on organismal biology, tumor suppression, and ageing/senescence. For example, we discuss in detail transgenic mouse models with extra copies of TP53, including the ‘super p53’ mice, with respect to potential trade-offs with traits such as premature aging and accelerated reproductive senescence. We also suggest that elephants may have avoided these costs because functional TP53RTG genes likely evolved through non-functional intermediates, which accumulated loss of function mutations that minimized redundancy with TP53.

As we discuss in greater detail above, our data combined with the data published in Abegglen et al. suggest that at least five TP53RTG genes are transcribed, many of which are likely-tissue specific. However, we did not mean to imply that all, or even most TP53RTGs are expressed and function. Indeed, we suspect that many are non-functional pseudogenes generated by a simple birth-and-death process in which the birth rate exceeds the death rate leading the neutral accumulation of TP53RTG pseudogenes. This process can be expected to “snowball” leading to the rapid accumulation of pseudogenes. To address this concern we have added a section to the Discussion (‘An embarrassment of riches?’) in which is explicitly state that we expect only a subset of TP53RTG genes will be functional at any point in the evolution of elephants and that the gene family is likely evolving by a birth and death process.

With respect to potential expressed or functional TP53 pseudogenes in mouse and rat, we have extensively surveyed the literature including Tanooka et al. (1998) and Tanooka et al. (2001) and cannot find evidence that Muroid TP53 pseudogenes are expressed. Tanooka et al. (1998), for example, report the formation of a chimera between TP53 and a TP53 pseudogene in mice that arises because of homologous recombination after repeated local β-irradiation. The recombination occurred near the 5' end of exon 5 leading to a 5-bp deletion in exon 6 of the expressed TP53 allele. A follow up study reports the generation of transgenic mice carrying a mutant version of this TP53 (mp53) allele, but neither study provides evidence that the pseudogene is expressed. Similarly, Tanooka et al. (2001) report the identification of the TP53 pseudogene in multiple populations and species of mice but do not provide evidence that these pseudogenes are expressed. We have identified several papers describing TP53 pseudogenes in mouse and rat (Ciotta et al., 1995; Czosnek et al., 1984; Hulla, 1992; Tanooka et al., 1995; Weghorst et al., 1995; Zakut-Houri et al., 1983), but none report expression data. Thus we conclude there is little direct evidence that TP53 pseudogenes are expressed in mouse and rat.